# Heterogeneous Graph Transformers for Simultaneous Mobile Multi-Agent Task Allocation and Scheduling under Temporal Constraints

**Batuhan Altundas***
Georgia Institute of Technology
Atlanta, GA 30332
baltundas3@gatech.edu

**Shengkang Chen***
Georgia Institute of Technology
Atlanta, GA 30332
schen754@gatech.edu

**Shivika Singh**
Georgia Institute of Technology
Atlanta, GA 30332
ssingh794@gatech.com

**Shivangi Deo**
Georgia Institute of Technology
Atlanta, GA 30332
sdeo33@gatech.edu

**Minwoo Cho**
Georgia Institute of Technology
Atlanta, GA 30332
mcho318@gatech.edu

**Matthew Gombolay**
Georgia Institute of Technology
Atlanta, GA 30332
matthew.gombolay@cc.gatech.edu

## Abstract

Coordinating large teams of heterogeneous mobile agents to perform complex tasks efficiently has scalability bottlenecks in feasible and optimal task scheduling, with critical applications in logistics, manufacturing, and disaster response. Existing task allocation and scheduling methods, including heuristics and optimization-based solvers, often fail to scale and overlook inter-task dependencies and agent heterogeneity. We propose a novel Simultaneous Decision-Making model for Heterogeneous Multi-Agent Task Allocation and Scheduling (HM-MATAS), built on a Residual Heterogeneous Graph Transformer with edge and node-level attention. Our model encodes agent capabilities, travel times, and temporospatial constraints into a rich graph representation and is trainable via reinforcement learning. Trained on small-scale problems (10 agents, 20 tasks), our model generalizes effectively to significantly larger scenarios (up to 40 agents and 200 tasks), enabling fast, one-shot task assignment and scheduling. Our simultaneous model outperforms classical heuristics by assigning 164.10% more feasible tasks given temporal constraints in 3.83% of the time, metaheuristics by 201.54% in 0.01% of the time and exact solver by 231.73% in 0.03% of the time, while achieving $20\times$-to-$250\times$ speedup from prior graph-based methods across scales.

## 1 Introduction

Multi-Agent Task Allocation and Scheduling (MATAS) seeks to determine the optimal assignment of tasks to agents, e.g., drones delivering supplies across disaster zones [1], and establish a corresponding schedule for each agent, accounting for both temporal and spatial constraints. Generation of optimal plans is essential in multi-agent systems, with applications ranging from search and rescue [2–5],

---

*These authors contributed equally to this work

39th Conference on Neural Information Processing Systems (NeurIPS 2025).

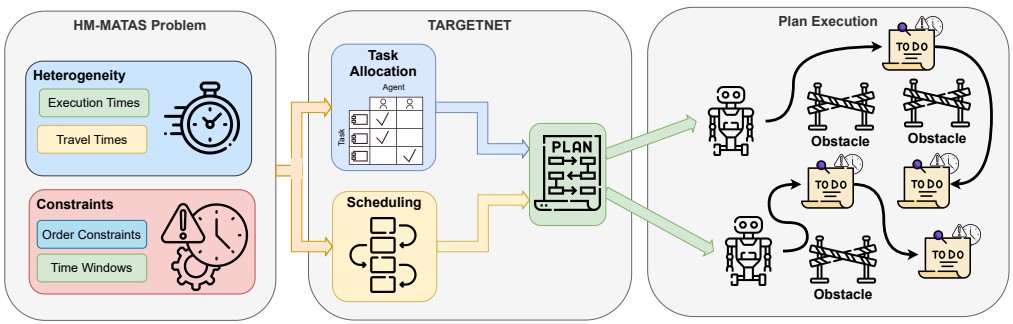

Figure 1: Simultaneous Task Allocation and Scheduling for Mobile Multi-Agent Teams with Heterogeneous Travel Time and Task Execution Time under Temporal Constraints. TARGETNET takes in observation and outputs task assignment and schedule. The environment executes and evaluates the optimality and feasibility.

warehouse automation [6, 7], transportation systems [8–11] and satellite coordination in space [12–14]. The simplest MATAS problem is the Multiple Travelling Salesman Problem (mTSP), optimizing the travel cost of multiple agents, making it an NP-hard problem that can become more complex with the addition of complex dependencies and constraints such as task coupling, time windows and heterogeneity in tasks and agents based on physical capabilities [15–24].

We aim to develop a generalized approach that integrates these two subproblems within a unified model, considering task and agent heterogeneity.

While constraint satisfaction methods can provide exact solutions to MATAS by handling temporal and spatial constraints [25, 26], search-based optimizers scale poorly with increases in the number of tasks, agents, and constraints, and require handcrafted encodings and hand-tuned parameters, making them brittle for new domains. Metaheuristics, like Genetic Algorithms (GA) [27–30], and market-based approaches [31, 32] offer alternatives but rely on domain-specific knowledge and can provide suboptimal results.

Recent Graph Neural Network (GNN)-based solvers [23, 24, 33–38] show that learning based approaches can provide near-optimal results while remaining scalable by learning from data instead of relying on hand-crafted solvers. However, GNNs face limitations in capturing dynamic, heterogeneous relations essential in MATAS for heterogeneous agents due to the method of aggregation messages from multiple sources. Attention-based methods have been proposed to address these limitations [39], which are further expanded in Heterogeneous Graph Transformers (HGT) [40, 41] leverage attention mechanisms that allow effective modeling of heterogeneous and pairwise dynamics, making them suitable for MATAS tasks with complex relations.

We tackle the challenge of Heterogeneous and Mobile MATAS (HM-MATAS) problems using scalable learning-based models. We introduce Task-Agent Relational Graph Encoding for Team-based Navigation and Execution of Tasks (TARGETNET), an approach that leverages a relational modeling framework that represents the Optimization Problem as a graph. TARGETNET assigns an agent to each task and schedules the tasks in the presence of order and time-window constraints. By leveraging relational representations, our method learns to generate closer-to-optimal policies than the baselines while allowing for deployment to larger-scale problems. TARGETNET generates schedules in a fraction of the time it takes for the optimizer to find optimal solutions larger scales. Our results show that leveraging the expressiveness of Graph Transformers with edge attention and residual networks can allow for the creation of a fast and scalable policy, delivering robust and fast scheduling solutions for real-world multi-agent systems.

In summary, these are the main contributions:

1. We present TARGETNET for modelling and solving HM-MATAS problems, enabling fast and scalable multi-agent task planning, allowing training on small instances and deployment on larger problems.

2. We propose a novel Node and Edge Attention-Aware Heterogeneous Graph Transformer with Residuals for improved performance in modeling heterogeneous and dynamic relations present in edge features. By leveraging the attention mechanism our method learns generalized policies that scale to larger problems without retraining.

3. We compare our model against state-of-the-art models, showing that our model is able to generate schedules that are 13.26% better than the best performing metaheuristics in 0.73% of the time, and 36.34% better performance than state-of-the-art GNN-based schedules in 4.75% the time in 10 agent-20 task problems, while generating 231.73% better than partial schedules generated by exact solvers after 12 hours in less than 12 seconds, in 40 agent-200 task problems.

## 2 Related Works

### 2.1 Multi-Agent Task Allocation

Gerkey and Mataric [15] proposed a taxonomy of the MRTA based on agent types (single-task vs. multi-task agents), task types (single-agent vs. multi-agent tasks), and assignment type (instantaneous assignment vs. time-extended assignment). In this work, we focus on single-task agents, single-agent tasks, and time-extend assignment (SR-ST-TA) problems which is a combinatorial optimization problem known to be strongly NP-hard. Unlike prior work [33, 24], we further account for Heterogeneous Travel Time of agents along with interrelated constraints. The location of the other agents and the tasks that they are able to complete influence the performance of other agents, leading to Complex-Schedule Dependencies (CD) [16].

#### 2.1.1 Non-learning-based Approaches

Researchers have developed different types of approaches to address the MATAS problem. Market-based approaches [31, 42, 43] and game-theoretical approaches [44, 45] are decentralized methods that enable distributed decision-making, but they typically yield worse performance compared to centralized approaches. Metaheuristics methods, including Ant Colony Optimization (ACO) [46, 47, 36] and Genetic Algorithm (GA) [30] rely on search-based heuristics without any optimality guarantees. Since the MATAS problem is an optimization problem, it can be represented as a Mixed Integer Linear Programming (MILP) problem [26], allowing for exact solvers [48, 49] to generate solutions. Chakraa et al. [50] provides a detailed survey of existing optimization-based methods. However, most of these optimization-based approaches are computationally expensive and scale poorly with problem size.

To handle complex time constraints in task allocation and scheduling, various search-based methods have been proposed. Nunes and Gini [26] developed an auction-based method considering time-window constraints. Whereas Kartal et al. [51] used Monte Carlo Tree Search (MCTS) to address MATAS with time-window constraints as well. Choudhury et al. [52] applied conflict-based search to find solutions under complex temporal constraints. However, these methods require expert domain knowledge to adapt to different problems, while our method of graph representation can encode the problem features and learn based on a given objective function through trial and error, making it more versatile for different objective functions.

#### 2.1.2 Learning-based Approaches

Learning-based MATAS methods model are function approximators [53] that trade-off training time to learn heuristic policies allow fast deployment without the need for experts [25]. Multi-Agent Reinforcement Learning (MARL) is used to train different models, such as Decision Trees [54], Neural Networks [55–57, 38, 58] and Recurrent Neural Networks (RNN) [55, 59]. The models were trained using different reinforcement learning algorithms, such as value-based [60, 61], policy-based [24, 33, 62], and actor-critic based [63, 64] learning. However, these approaches are limited in their modeling capabilities, only addressing a subset of MATAS problems, such as Vehicle Routing Problems [65, 66], multiple Travelling Salesman Problems [67, 68], homogeneous agents, no travel time [33, 24], and struggling to scale effectively to larger, more complex environments or providing interpretability [69].

### 2.2 Graph Neural Networks

GNNs provide a scalable approach to solving problems in multi-agent coordination [35, 70–72], adversarial agent modelling [73], and human-agent teaming [74]. Wang and Gombolay [33] proposed

a new type of Heterogeneous Graph Attention Networks (HetGAT) [75] with Edge Features [76] representing heterogeneous relational information for multi-agent task scheduling problems by sequentially assigning agents to tasks. Transformer Attention mechanism has been used to solve limited scale constraint satisfaction problems such as Sudoku, however these methods do not scale up to larger problem sizes, and require a dense attention matrix to represent the constraints [77]. Heterogeneous Graph Transformers (HGTs) have been shown to outperform other attention based graph models in optimization problems while allowing for scalable and sparse relational representation [40], and they have been used for the Task Allocation for mTSP problems, followed by using an optimizer for solving the scheduling per agent without temporal constraints [62, 68]. However, existing HGT architectures lacks the ability to process relational information encoded within edge features. Our work leverages the capabilities of Heterogeneous Graph Transformers (HGTs) to address MATAS problems, effectively accounting for heterogeneous travel times and task completion durations.

# 3 Methodology

In Section 3.1 we present the HM-MATAS problem as an Optimization Problem, and in Section 3.2 present a Markov Decision Process (MDP) to simulate the movement of agents as they complete the tasks, allowing for a framework that can be used to train learning-based models using Reinforcement Learning [78]. In Section 3.3 we present a method of encoding the CSP problem in a graph framework that can be adapted for single-step Task Allocation and Scheduling. In Section 3.4 we present a new method of integrating Edge Features into GNNs, which we combine with the graph representation and train to solve HM-MATAS problems.

## 3.1 Problem Setup

**Task Allocation and Scheduling under Temporal Constraints:** We consider HM-MATAS problem for mobile team of agents $\mathcal{A} = \{1, 2, \ldots |A|\}$ and a set of tasks $\mathcal{T} = \{1, 2, \ldots, |T|\}$. Agents start in random locations and must travel to the location of the task to start the execution of the task. The heterogeneous execution time is the time it takes for agent to start a task $j$, $t_{ij}^S$ and finish it, $t_{ij}^F$, presented as $t_{ij}^E = t_{ij}^F - t_{ij}^S$. We also account for order constraints, where $W_{jk}$ is the minimum required wait-time between the completion of task $j$ and start of task $k$ and every task has a time window, $[s_k, e_k]$, where $s_k$ is the earliest start time, and end before the deadline constraint, $e_k$. The objective function is to minimize the maximum time it takes to complete all tasks, denoted as the makespan, while satisfying the temporal constraints. The MILP representation of the Optimization Problem can be found in the Appendix A.

**Heterogeneous Trave Times using Precomputed Motion Plan:** Physical heterogeneity of agents include heterogeneous velocities which impact the travel time from one task to another [22]. We use Rapidly-exploring Random Graph Algorithm [79] to compute the travel distances between agent start locations to task locations and the locations of pairs of tasks in the presence of obstacles. The heterogeneous velocities of each agent $i$, lead to having distinct travel times along the same path, represented as $t_{i0j}^T$, for travel time from initial location to task $j$, and $t_{ijk}^T$ for travel time of agent $i$, from the location of task $j$, to task $k$. Abstracting the motion-planning enables integration with diverse planners [80–82], while preserving downstream optimization tractability [83].

## 3.2 Markov Decision Process

We present the HM-MATAS problem in the form of a five-tuple Markov Decision Process $< S, A, P, R, \gamma >$ that simulates the movement of agents through the map to complete the assigned tasks in a given ordered schedule. The **state space**, $S$, captures the state of the environment based on the current location of agents and tasks, when the agents will be available, a list of assigned and unassigned tasks, along with temporal constraints. The **action space**, $A$, consists of all possible agent-task pair selections, where each single action at time step $t \in [1, |T|]$, $a_t = \langle \alpha_i, \tau_j \rangle$, represents assigning agent, $\alpha_i$, to task, $\tau_j$. Each task is only assigned to an agent once, limiting the time horizon to number of tasks, $|T|$. The **Transition Function**, $P$, updates the state of the model, moving agents to new assigned task locations and generates a new graph representation after assignment. The **reward function**, $R$, maximizes the number of feasible decisions, represented by a boolean function $\mathbb{1}_{\text{feasible}}(s_t, a_t)$, at every time-step, $t$, while minimizing the maximum makespan, $t_{ms}$, normalized

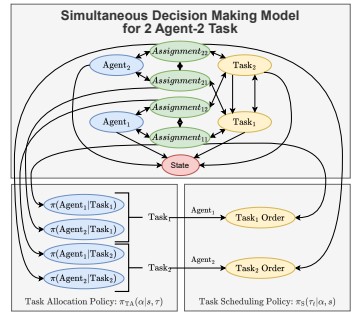

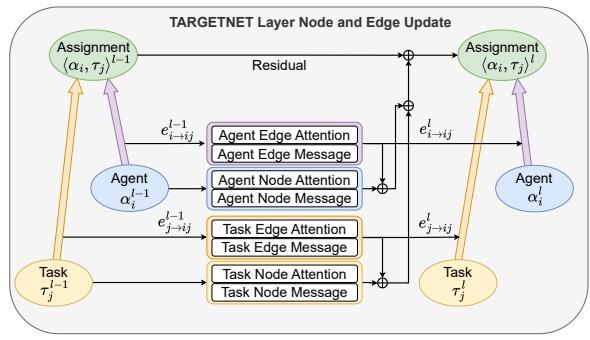

(a) Sample graph representation of a simple 2 agent-2 task problem, showing the heterogeneous nodes, and relationships for Task Allocation and Scheduling using TARGETNET.

(b) Graph Aggregation using Edge and Node Attention and Message Passing with Residuals. Assignment node is updated using the combined Messages from agent and task nodes and edges based on Attention.

Figure 2: Graph-based Task Allocation and Scheduling using GNNs with node and edge features.

to maximum deadline time, $t_{ddl}$. The constraint feasibility and optimality are combined using hyperparameters $\alpha$ and $\beta$, for a total reward as shown in Eq. 1. To balance the short-term feasibility with long-term plan quality and clarify credit assignment, we combine dense feasibility reward, $+1$ if feasible, $-1$ otherwise, with sparse reward at final timestep, $t = |T|$, that combines number and optimality of feasible assignments as shown in Eq. 1. $\gamma \leq 1$ is the **discount factor**.

$$R(s_t, a_t) = \begin{cases} \alpha(1 - \frac{t_{ms}}{t_{ddl}}) + \beta \sum_{i=1}^{T} \mathbb{1}_{\text{feasible}}(s_i, a_i)) & , \text{if } t = |T| \\ +1 & , \text{if } \mathbb{1}_{\text{feasible}}(s_t, a_t) \\ -1 & , \text{otherwise} \end{cases} \quad (1)$$

### 3.3 Graph Modelling of Heterogeneous Task Allocation and Scheduling with Travel Times

We represent the HM-MATAS problem as a heterogeneous graph model $G = (V, E)$ with vertices $V$ to describe *Agent*, *Task*, *Assignment* of an agent to task, global *State* and *Task Order* nodes. Heterogeneous edges $E$ that encode the relations between different heterogeneous elements of the graph. Each heterogeneous node encodes information unique to the node type, such as velocity and earliest available time for agents and time window constraints for tasks. Edges encode relational information between the two node types, representing travel time from agent's current location to tasks, and travel time and distance between tasks. We further augment our graph representation with the *Assignment* node to simulate the assignment of an agent to a task and to explicitly model heterogeneous task execution time [84]. The *Assignment* nodes representing a specific agent have edges between each other to encode the heterogeneous travel time from one task to another for the specific agent. Consequently, there are $O(|A||T|)$ *Assignment* nodes in the graph for $|A|$ agents and $|T|$ tasks, with $O(|A||T|^2)$ edges between the assignment nodes in total. Full details of the node and edge features can be found in Appendix B.1.

### 3.4 Graph Transformer Network with Edge Attention

GNNs were used to some MATAS problems represented as graphs [33, 61, 62, 68] with Attention mechanisms [39, 40] showing higher performance by accounting for the relational importance of each edge. HGTs outperform other attention-based graph networks by leveraging the Transformer Attention mechanism [40, 85] in standard benchmarks. However, HGTs support node-type-specific attention, but does not innately support edge features, which are crucial for MATAS [77].

We augment each message-passing step in HGTs, as described in Appendix B, with edge-based attention and messaging leveraging the distributive property of attention [86]. The node and edge features combine to update of target node feature, $h^l[t]$, and edge features, $h^l[e]$, for the next layer, $l$, for an edge, $e$, connecting source node, $s$, to target node, $t$, as shown in Eq. 2 and 3 respectively.

$$h^l[t] \leftarrow \underset{\forall s \in N(t) \forall e \in E(s,t)}{\textbf{Aggregate}} \left( \text{Att}_N(s,t) \cdot \text{Msg}_N(s) + \text{Att}_E(e,t) \cdot \text{Msg}_E(e) \right) \tag{2}$$

$$h^l[e] \leftarrow \text{Att}_E(e,t) \cdot \text{Msg}_E(e) \tag{3}$$

We pass information from edge features to target nodes using message passing. For multi-head attention, we use $M_E^i(e) = W_{M_E}^i h_e^{l-1}$, to calculate message vectors for head $i$. $W_{M_E}^i$ and $W_{\phi(e)}^{MSG}$, along with the standard HGT weights [40] described in Appendix B, are trainable.

$$\text{Msg}_{\text{E}}(e,t) = \left\|_{i=1}^{h} M_E^i(e) W_{\phi(e)}^{MSG} \right. \tag{4}$$

The edge attention, Eq. 5, determines how important each edge message is to the target node, using edge features, $h_e^{l-1}$, and target features, $h_t^{l-1}$. The key values of the edge, $K_E^i(s) = W_{K_E}^i h_e^{l-1}$, and query values for target node, $Q_E^i(t) = W_{Q_E}^i h_t^{l-1}$, are used to compute the attention, normalized using the vector dimension of $d$, and the learned prior tensor $\mu_{\langle \phi(e), \tau(t) \rangle}^N$. The multi-head attention allows for the learning of multiple weights for the same features that are combined to increase the descriptive power of attention [85]. While the standard attention mechanism applies a softmax across multi-head attention, this formulation enforces that the policy always accounts for both edge and node features, even when some policies may not require them. By omitting the softmax, we allow attention weights to reach zero, enabling the model to learn policies that selectively ignore certain components of the graph. This flexibility improves the representational capacity of the model [87]. Key, query and attention weights, $W_{K_E}^i$, $W_{Q_E}^i$ and $W_{\phi(e)}^{\text{ATT}}$ respectively, are learned and used to compute the attention used to update both edge and target features.

$$\text{Att}_E(e,t) = \left\|_{i=1}^{h} K_E^i(s) W_{\phi(e)}^{\text{ATT}} Q_E^i(t)^T \right\rangle \frac{\mu_{\langle \phi(e), \tau(t) \rangle}^N}{\sqrt{d}} \tag{5}$$

By accounting for both node and edge features, we present a GNN algorithm that can be used for problems where relational information is vital for decision making, such as MATAS problems.

**Residual Graph Neural Networks:**   We adopt a specific type of residual network based upon the graph-raw residual [41] concept, where the initial input of the graph node and edge features are appended to each layer, to mitigate vanishing gradients in deep GNNs [88].

### 3.5   Simultaneous Decision Making Policy Training

We process the input features, represented as a graph model in Section 3.3, using the edge-feature augmented HGT, as described in Section 3.4, to generate output predictions for Task Allocation and Scheduling policies. Our model takes in the initial state, $s_0$, and generates a Task Allocation, $TA$, and Task Scheduling, $Sch$, without iteratively interacting with the environment, allowing a significant boost to computation speed.

**Task Allocation:**   We leverage the output features of the *Assignment* nodes, which represent the pairwise assignment of $|A|$ agents to $|T|$ tasks with a total of $|A||T|$ nodes. The *Assignment* node features of the output is reshaped into a $|A| \times |T|$ matrix, and a single agent is sampled for each task to achieve Task Assignment for single-agent tasks.

**Scheduling:**   We sort the Tasks using the output of the *Task Order* nodes connected to each task. During training, scheduling is performed by sampling tasks sequentially from a Probability Mass Function over Task Order node outputs. Sampling proceeds without recomputing weights, requiring only a single GNN forward pass.

**Training:**   Our policy computes the Task Allocation and Scheduling in a single forward pass. However, we leverage the sequential nature of scheduling to train our models using REINFORCE [78, 89, 90] with log probabilities based on prior work for numerical stability [24]. We use the discounted reward, combining sparse final performance for optimization and dense feedback on individual task

constraint satisfaction. At any given time-step, $t$, the Task Assignment policy, $\pi_{TA}$, assigns an agent, $\alpha_t$, to a task, $\tau_t \in T$. Simultaneously, the Scheduling policy, $\pi_{Sch}$, determines which task, $\tau_t$, occupies the $t^{th}$ position in the plan. This scheduling decision is conditioned on the history of assigned tasks, $\tau_{<t}$, , the complete set of agent-task pairings, $\langle \alpha, \tau \rangle_{all}$, and the initial state, $s_0$. We compute the gradient of the model using Eq. 6:

$$\nabla_\theta J(\theta) = \mathbb{E}_\pi \sum_{t \in 1:|T|} \left( \sum_{i \in t:|T|} \gamma^{|T|-i} R_i(\langle \alpha_t, \tau_t \rangle, s_0) \nabla_\theta \left( \log \pi_{TA}(\alpha_i | \tau_i, s_0) + \log \pi_{Sch}(\tau_t | \tau_{<t}, \langle \alpha, \tau \rangle_{all}, s_0) \right) \right) \quad (6)$$

## 4 Experiments

### 4.1 Experiment Setup

We generate our problems using the constraints described in Section 3.1, validated to have an optimal solution using exact solvers based on the MILP described in Appendix A. All learning-based policies are trained on 200 small-scale problems (10 agents-20 tasks), with the configuration outlined in Table 1 of Appendix E. Three instances are initialized per model with different random seeds and trained in parallel. For testing, 200 problems are generated for small and medium scales (10 agents-50 tasks), 30 for large scale (20 agents-100 tasks), and 10 for extra-large scale (40 agents-200 tasks). Due to the complexity of large and extra-large-scale problems, we generate 2 and 4 medium-scale problems, respectively, and combine them into a single map with shared obstacles. For each learning-based model, we train three instances with different random seeds and, for each test problem, report the performance of the best-performing instance. We then compute the mean and standard deviation of these best-case results across the problem set. Time performance is calculated by solving 10 problems of the given scale using the models. Exact details of problem generation and reproducibility of our experiments can be found in Appendix E.

The following metrics are used to evaluate the performance of the policy. These capture quality (optimality), feasibility, and practical utility (training and inference speed):

- **Optimality Rate:** The Optimality Rate of the Final Reward is acquired using the reward for the total schedule at time $|T|$, as described in Eq. 1, normalized to the reward of the MILP Solver, $O^\pi = R^\pi_{|T|}/R^{MILP}_{|T|}$.

- **Feasibility Percentage:** The Percentage of Feasible Task Assignments is acquired by normalizing the number of feasible assignments to the total number of tasks for each problem, $|T_{feasible}|/|T|$.

- **Training Speed:** The mean training time per episode using a training dataset using small scale problems, relevant for learning-based models.

- **Inference Speed:** Mean wall-clock time it takes to generate the schedule in seconds.

#### 4.1.1 Models and Baselines

We compare **TARGETNET** using Edge Enhanced HGT with Residuals against Heuristic and Metaheuristic schedulers, and compare the model performance against existing Graph-based Learning Models, that leverage the Task Allocation Graph Representation Model presented in Fig. 7.

**MILP Solver:** Exact solver using the MILP formulation described in Appendix A to solve the problem [48].

**Heuristics:** We compare our model against greedy heuristic solvers, **Earliest Deadline First (EDF)** that prioritizes tasks with earlier deadlines and assigns them to agents with earlier completion times [91] and **Constraint-Aware EDF (CA-EDF)** that accounts for precedence in task order dependencies [26, 51].

**Metaheuristics:** A metaheuristic that improves a population of schedules over several generations by applying mutations and selecting the best schedules based on an initial seed [30, 29]. We utilize Genetic Algorithm to improve the performance of a starting scheduling generated randomly, **Gen-Random**, and generated using EDF heuristic, **Gen-EDF**. Given the high time complexity of the Genetic Algorithm, we limit the evaluation to the best-performing schedules after 1 and 3 generations

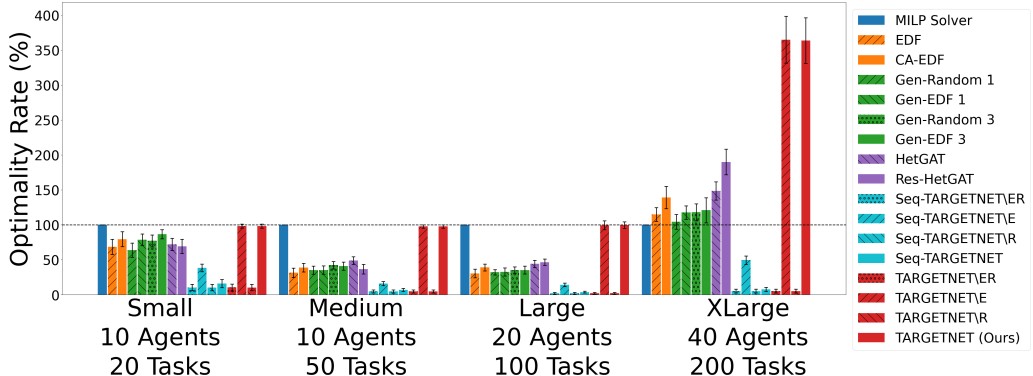

Figure 3: Mean optimality rate of the final reward on different scales, with standard deviation as error bars. Higher is better.

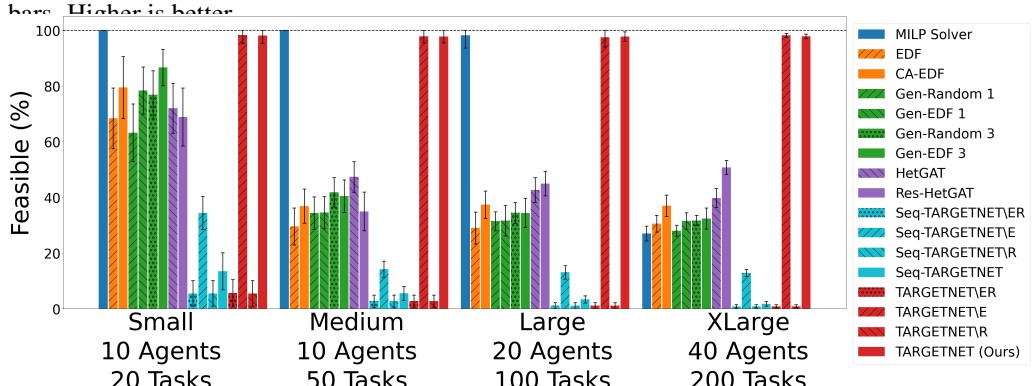

Figure 4: Mean percentage of feasible task Assignments on different scales, with standard deviation as error bars. In extra-large scales, graph-based models are shown to be able to outperform the optimizer. Higher is better.

of mutations based on computation time to be comparable to the optimal solver, with a population size of 100 and survival rate of 10% for the next generation.

**GNN-Baselines:** We evaluate GNN-based models that take in the Graph Model described in Section 3.3 as input to generate sequential agent-task assignment policies as described in prior work [24]. Our baselines include **HetGAT** [61], which employs graph-attention mechanism with edge-level attention for sequential decision making, and its residual-enhanced variant, **Res-HetGAT**, based on the graph-raw-residuals [88]. Additionally, we benchmark our Simultaneous Decision Making model against a Sequential Decision Making variant, referred to as **Seq-TARGETNET**. We also consider two simplified versions: **Seq-TARGETNET\R**, which excludes residual connections, **Seq-TARGETNET\E**, which excludes edge features and **Seq-TARGETNET\ER**, which omits both edge features and residuals.

**Ablation:** To isolate the contributions of residual connections and edge features, we conduct an ablation study on our simultaneous decision-making model. Specifically, we evaluate **TARGETNET\R**, which removes residuals, **TARGETNET\E** which omits edge features and **TARGETNET\ER**, which removes both residuals and edge features, allowing us to assess their individual and combined impacts on performance.

## 5 Results & Experiment Analysis

We evaluate the performance of our proposed TARGETNET model across a range of problem sizes, comparing it with rule-based heuristics, metaheuristics, GNN-based sequential policies, and an MILP-based optimal solver. Our primary focus is on the trade-off between computational efficiency and solution quality, specifically feasibility and optimality of task assignments.

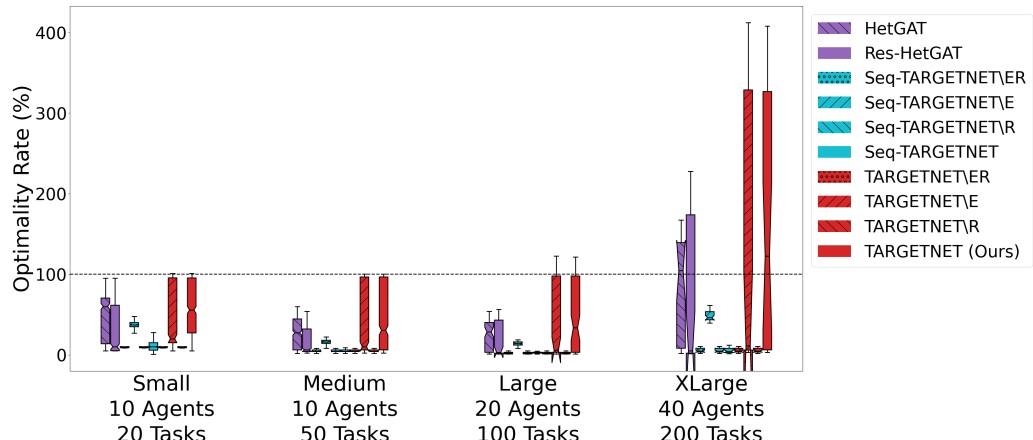

Figure 5: Mean optimality rate of the final reward for 3 seeds of the learning-based models on different scales, with standard deviation as error bars. Higher is better.

TARGETNET consistently delivers near-optimal performance while significantly reducing computation time. In small-scale problems, it achieves a 13.27% improvement in feasible task assignments over the rule-based CA-EDF policy and 36.35% improvement over HetGAT [61], generating schedules in under 0.20 seconds, the only model besides heuristics capable of sub-second scheduling at small scale.

As problem size increases, the computational benefits of TARGETNET become more pronounced. In extra-large scale settings, where the MILP solver fails to return fully-feasible solutions within 12 hours (producing partial schedules in all 10 cases), TARGETNET generates schedules with 70.80% more feasible tasks than the MILP solver and 47.05% more than HetGAT in 0.37% the computation time as while sequential models benefit from per-decision re-optimization but suffer cubic complexity, taking $538.56\times$ more time to generate schedules. While TARGETNET\E achieves similar performance to TARGETNET for the best performing seed, on average across 3 seeds, TARGETNET performs from 16.91% to 34.75% across different problem scales.

The inductive generalization ability of our graph-based encoding combined with simultaneous Task Allocation and Scheduling of TARGETNET allows our model to be trained on small-scale instances, and deployed directly to significantly larger problem sizes without retraining, still maintaining competitive performance. As shown in Fig. 5, TARGETNET consistently outperforms both baseline methods and ablation variants across all three random seeds. Although the TARGETNET\E variant achieves performance comparable to the full model in one seed (seed 10), the overall results demonstrate that TARGETNET maintains superior robustness by effectively representing policies both with and without explicit edge feature conditioning through the attention mechanism. Our empirical results suggest that incorporating edge-level information contributes to more stable generalization and reduced brittleness across varied problem configurations.

From a computational complexity perspective, TARGETNET operates with a time and space complexity of $O(|A||T|^2)$, for number of agents, $|A|$, and number of tasks, $|T|$. This contrasts with the $O(|A||T|^3)$ complexity of sequential models such as HetGAT and Seq-TARGETNET leveraging the same graph structure, which require repeated environment updates and forward passes for each decision step. Our single-step schedule generation framework allows for a significant reduction in computational load, making TARGETNET suitable for real-time deployment scenarios where optimal solvers and sequential GNNs become impractical.

The training time for the Simultaneous model yields up to 20x improvement over sequential methods per episode as shown in Fig. 6b. Furthermore, while the model is trained only on the small scale, empirical results show that training in small scale is sufficient enough to learn policies that can generalize to larger problem sizes.

Empirical timing results shown in Fig.6b confirm the advantages highlighted in computational complexity. TARGETNET is consistently faster than all non-heuristic baseline methods across scales and remains the only graph-based model capable of combining low-latency inference with near-optimal task assignment performance. In medium to extra-large scales, TARGETNET is 3.90 to 26.18 times faster than heuristics.

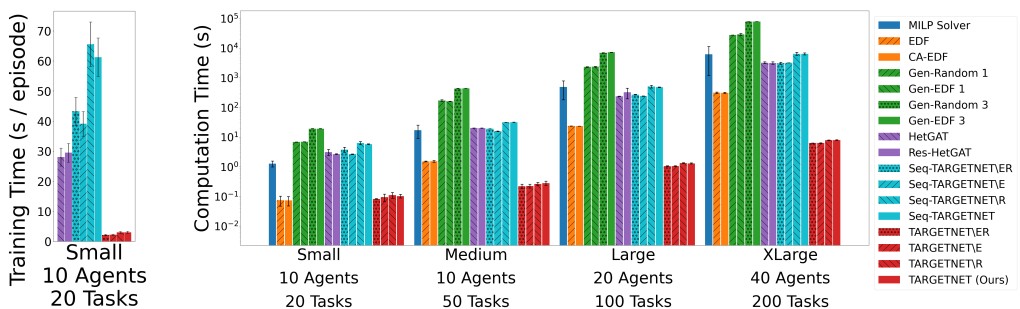

Figure 6: (a) Mean Training Time per episode and (b) Mean Computation Time during Testing, with standard deviation as error bars. Lower is better.

TARGETNET achieves comparable or better performance while maintaining orders-of-magnitude faster runtime, offering a $20\times–250\times$ speedup over state-of-the-art sequential GNN-based models. These results underscore its capacity to produce high-quality schedules under tight runtime constraints.

We further conduct a sensitivity analysis on a subset of constraints and agent heterogeneity that our policies are trained on, and analyze the training stability across models initialized with three random seeds three in Appendix G.

## 6 Limitations and Future Work

One of the key limitations of solving mobile task allocation and scheduling problems with pre-computed motion plans is that represented model of the problem does not account for a dynamic environment and agent-to-agent collision avoidance. The forecasting travel time in our experiments assumes static path planning, which may lead to inaccuracies in scheduling when faced with unexpected changes in the environment. Future work should account for agent-to-agent collision during multi-agent path finding to allow for higher fidelity solutions in real world applications [38, 83].

Our experiments show that simultaneous methods consistently outperform non-exact solvers and run faster across scales, while sequential models yield superior results at larger scales but are significantly slower. We propose a hybrid sequential–simultaneous architecture that merges the speed of simultaneous models with the scalability of sequential approaches. By integrating both through a metaheuristic that selects the best output based on performance and computation time, we deliver a robust, scalable solution for optimization and constraint satisfaction. Furthermore, TARGETNET provides schedules that can be used to warm-start the optimizer, reducing the time that the optimizer takes to return an exact solution by providing a better solution [92].

While TARGETNET learns a scalable policy, the graph-based models are not inherently interpretable, and the learned policy is hard to validate by humans [93]. While existing graph-based explainability methods, such as saliency maps or counterfactuals [93], provide insight into learned policies, a graph model that has full-transparency in its structure [94] is essential for safety-critical MATAS.

## 7 Conclusion

In this paper, we present TARGETNET, a scalable, generalizable, and one-shot framework for Multi-Agent Task Allocation and Scheduling (MATAS) that integrates relational graph reasoning through Heterogeneous Graph Transformers with edge-specific attention. By capturing dynamic agent-task relationships and modeling the combinatorial nature of scheduling problems via a novel graph representation, TARGETNET enables scalable simultaneous decision-making, learning policies that enable inference an order of magnitude faster than graph-based sequential solvers. Our approach outperforms both heuristic baselines and genetic metaheuristics across problem scales—from small to extra-large—while remaining computationally efficient and achieving near-optimal performance in scenarios where exact solvers fail. We show that TARGETNET learns high-performing generalizable policies that break the trade-off between speed, accuracy, and scalability.

## Acknowledgments and Disclosure of Funding

This work was sponsored by Naval Research Laboratory (NRL) under grant numbers N00173-21-1-G009 and N00173-25-1-0050, and by Office of Naval Research (ONR) under grant number N00014-23-1-2887

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

# Appendix A   Mixed Integer Linear Program Solver for Task Allocation and Scheduling

The Heterogeneous Mobile Multi-Agent Task Allocation and Scheduling Problem can be formulated as a Mixed Integer Linear Program (MILP). This formulation enables the joint optimization of task assignments and their execution schedule.

We adopt the MILP formulation presented in Eq. 7, where the objective is to optimize the schedule according to a predefined cost function. We build on prior work [16, 25], with the key extension of incorporating heterogeneous travel times. These travel times are derived from varying agent velocities and precomputed distances between the agents' initial positions and task locations.

$$
\begin{aligned}
\min \quad & f(\mathbf{A}, \mathbf{S}^1, \ldots, \mathbf{S}^{N_A}) \\
\text{s.t.} \quad & \sum_{i \in \mathcal{A}} \mathbf{A}_{ij} = 1 && \forall j \in \mathcal{T} && \text{(C1)} \\
& \mathbf{S}^i_{jk} + \mathbf{S}^i_{kj} \leq M(2 - \mathbf{A}_{ij} - A_{ik}) + 1 && \forall j, k \in \mathcal{T}, i \in \mathcal{A} && \text{(C2)} \\
& \mathbf{S}^i_{jk} + \mathbf{S}^i_{kj} \geq -M(2 - \mathbf{A}_{ij} - A_{ik}) + 1 && \forall j, k \in \mathcal{T}, i \in \mathcal{A} && \text{(C3)} \\
& t^A_k \geq -M\left(3 - (\mathbf{A}_{ij} + \mathbf{A}_{ik} + \mathbf{S}^i_{jk})\right) + t^F_j + t^T_{ijk} && \forall j, k \in \mathcal{T}, i \in \mathcal{A} && \text{(C4)} \\
& t^A_k \geq -M(1 - \mathbf{A}_{ik}) + t^T_{ik} && \forall k \in \mathcal{T}, i \in \mathcal{A} && \text{(C5)} \\
& t^S_k \geq t^A_k && \forall k \in \mathcal{T} && \text{(C6)} \\
& t^S_k \geq M(\mathbf{O}_{jk} - 1) + t^F_j + W_{jk} && \forall j, k \in \mathcal{T} && \text{(C7)} \\
& t^S_k \geq s_k && \forall k \in \mathcal{T} && \text{(C8)} \\
& t^F_k \geq t^S_k + \mathbf{A}_{ik} t^E_{ik} && \forall k \in \mathcal{T}, i \in \mathcal{A} && \text{(C9)} \\
& t^F_k \leq e_k && \forall k \in \mathcal{T} && \text{(C10)}
\end{aligned}
\tag{7}
$$

In this formulation, $M$ denotes a large constant used to model conditional constraints. The constraints are interpreted as follows:

- (C1) ensures that each task is assigned to exactly one agent.

- (C2) and (C3) enforce mutual exclusivity and sequencing when two tasks $j$ and $k$ are assigned to the same agent $i$.

- (C4) sets the arrival time, $t^A_k$, of an agent, $i$, to task, $k$, ensuring it occurs only after completing the preceding task, $j$, and arriving in the location of task, $k$, after travel time, $t^T_{ijk}$.

- (C5) ensures that the agent's arrival at its first assigned task considers its initial location.

- (C6)–(C8) ensure that task start times respect agent arrival times, required wait times between dependent tasks, and task time windows.

- (C9) accounts for heterogeneous task durations depending on the assigned agent.

- (C10) enforces task completion within the designated time window.

The objective function is defined as:

$$
f(A, S^1, ..., S^{N_A}) = \max_{j \in \mathcal{T}} \left( t^F_j \right)
\tag{8}
$$

which minimizes the makespan, i.e., the maximum task finish time across all tasks.

## Appendix B    Graph Neural Networks

### B.1    Node Attention and Message Passing in Heterogeneous Graph Networks

Hu et al. [40] describes the attention and message passing mechanisms for the nodes based on their relations. For each node, we compute key and query projections specific to attention head $i$ as follows:

$$K_N^i(s) = W_{K_N}^i h_s^{l-1}, \quad Q_N^i(t) = W_{Q_N}^i h_t^{l-1}$$

where $h_s^{l-1}$ and $h_t^{l-1}$ denote the input features of source node $s$ and target node $t$ from the previous layer, respectively. The matrices $W_{K_N}^i$ and $W_{Q_N}^i$ are learned projections for the $i$-th attention head.

The multi-head attention from source node $s$ to target node $t$ is computed as:

$$\text{Att}_N(s,t) = \left\|_{i=1}^{h} K_N^i(s) W_{\tau(s)}^{\text{ATT}} Q_N^i(t)^T \cdot \frac{\mu_{\langle \tau(s),\tau(t)\rangle}^N}{\sqrt{d}}\right. \tag{9}$$

where $\tau(s)$ and $\tau(t)$ denote the node types of $s$ and $t$, $W_{\tau(s)}^{\text{ATT}}$ is a type-specific transformation matrix, and $\mu_{\langle \tau(s),\tau(t)\rangle}^N$ is a learned scaling factor modulating attention between node types.

The corresponding message passed from node $s$ to $t$ is computed as:

$$\text{Msg}_N(s,t) = \left\|_{i=1}^{h} M_N^i(s) W_{\tau(s)}^{\text{MSG}}\right. \tag{10}$$

where $M_N^i(s)$ is the value vector for node $s$ in head $i$, and $W_{\tau(s)}^{\text{MSG}}$ is the type-specific transformation for message projection.

Graph Features

**Node Features:**    In the graph representation, the *Agent* node includes features such as Earliest Time Available, indicating the soonest the agent can begin a task, and Number of Tasks Assigned, reflecting current workload. The *Task* nodes contains temporal constraints and scheduling information, including Start Time Constraint, End Time Constraint, Expected Completion Time if assigned previously, and a binary feature indicating whether the Agent is Assigned. The *State* node captures high-level planning metrics such as the Number of Tasks Assigned, Maximum Makespan, the current Assignment state, and total Duration of scheduled activities.

**Edge Features:**    The relationships and transitions between nodes are encoded in the Edge feature. Each edge connection a source node to target node is represented as $\langle Source, Edge, Target\rangle$ and annotated with task-relevant attributes. The edge $\langle Agent, Duration, Task\rangle$ captures task execution time, while $\langle Agent, Travel\,Time, Task\rangle$ represents the movement cost. Inter-task dependencies are modeled with $\langle Task, Distance, Task\rangle$ and $\langle Task, Wait\,Time, Task\rangle$, encoding spatial separation and required wait periods, respectively. Assignment-specific relations are defined by $\langle Agent, Travel\,Time, Assignment\rangle$ and $\langle Agent, Task\,Duration, Assignment\rangle$, which record travel and execution times per assignment. Finally, inter-assignment transitions are denoted by $\langle Assignment, Travel\,Time, Assignment\rangle$, representing temporal costs between consecutive assignments.

## Appendix C    Sequential Decision Making Models

In the sequential task allocation setting, decision-making unfolds over a series of timesteps, each corresponding to the assignment of a single agent to a task. We employ a two-step actor mechanism with dedicated output nodes for *Agent Selection* and *Task Selection*, as seen in Fig. 8. The state of the environment is encoded as a graph and processed through a Graph Neural Network (GNN). The outputs are passed through *softmax* functions to produce probability mass functions (PMFs).

The agent selection policy, $\pi(\alpha_t|s_t)$, first selects an agent, $\alpha_t$, from the state, $s_t$, at time step, $t$. Subsequently, a task, $\tau_t$, is assigned to agent, $\alpha_t$, using the conditional policy $\pi(\tau_t|\alpha_t, s_t)$. Each task is assigned exactly once and removed from the pool of unassigned tasks upon completion. The joint action $a_t = \langle \alpha_t, \tau_t\rangle$ updates the environment to yield the next state $s_{t+1}$.

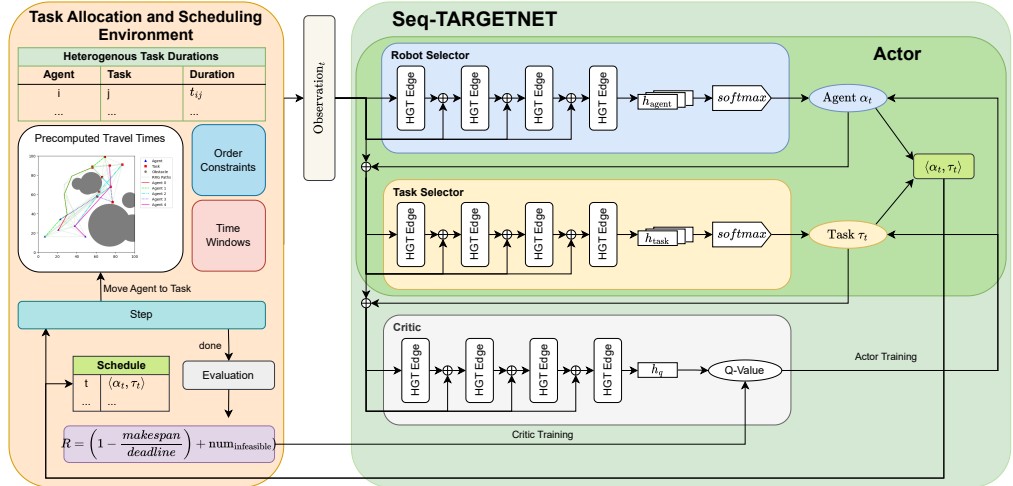

Figure 7: The training and policy pipeline for a multi-agent task allocation environment with obstacles and sequential decision-making using the Seq-TARGETNET baseline. The architecture comprises agent and task selection modules followed by a graph-based critic that evaluates each assignment. This pipeline allows for ordered assignment of agents to tasks and simulates their movement on a pre-computed motion planning map. Seq-TARGETNET uses a 4-layer Graph Neural Network with residual connections to enable robust sequential reasoning.

Unlike simultaneous assignment methods (e.g., TARGETNET), where assignments are computed from a single initial observation, the sequential formulation requires updating the environment iteratively after each decision. This enables more granular evaluation and dynamic response to partial assignments at the cost of added computational complexity [40, 24].

## C.1  Graph-based Critic for Sequential Training

Sequential Multi-Agent Task Allocation and Scheduling (MATAS) problems are characterized by sparse and delayed rewards, creating significant challenges for effective policy learning due to the well-known credit assignment problem [95]. In sequential settings, the decisions made by agents unfold over time, with each decision potentially influencing future states and the overall outcome of the schedule. The temporal dependency in sequential policies makes it nontrivial to attribute the eventual success or failure of the complete schedule to individual agent-task decisions. While the combined reward function defined in Eq. 1 incorporates both dense penalties for constraint violations and a sparse terminal reward for overall performance, the feedback remains insufficient for learning optimal intermediate actions in a long-horizon task.

In our framework, we adopt REINFORCE [89] for Simultaneous Task Allocation and Soft Actor-Critic (SAC) [96] for Sequential Task Allocation. Simultaneous task allocation presents a single-shot, static decision process where the complete task-agent assignment and ordering are generated in one step and evaluated holistically. REINFORCE is particularly effective in single-shot decision making, as it directly optimizes the expected return from full allocation decisions, using Monte Carlo estimates without requiring a value function or temporal credit propagation. Conversely, sequential allocation involves temporally extended decision-making with interdependent assignments and sparse, delayed rewards, making credit assignment a critical challenge [95]. To address this, we employ SAC, an off-policy actor-critic method that leverages value estimation and entropy-regularized learning to balance exploration and exploitation across long-horizon trajectories. This division enables each component to exploit the inductive biases of its underlying problem structure, leading to improved sample efficiency, learning stability, and policy performance [89, 96, 78].

The critic is trained to approximate the state-action value function, $Q_\theta(s_t, a_t)$, which estimates the expected cumulative return when executing action, $a_t$, in state $s_t$ and following the current policy thereafter. Its parameters $\theta$ are optimized by minimizing the squared error between the predicted

Q-value and the empirical return observed over the remainder of the episode. Formally, the critic loss is defined as:

$$\min_\theta J_Q(\theta) = \mathbb{E}_{(s_t, a_t) \sim \mathcal{D}} \left( \frac{1}{2} \left[ Q_\theta(s_t, a_t) - \sum_{i=t}^{|T|} \gamma^{|T|-i} R_i(s_i, a_i) \right]^2 \right) \quad (11)$$

where $\mathcal{D}$ is the replay buffer containing sampled trajectories, and $R_i(s_i, a_i)$ denotes the reward obtained at time step, $i$. The target return $\sum_{i=t}^{|T|} R_i(s_i, a_i)$ captures the total discounted return from timestep tt onward. This training encourages the critic to align its estimates with the actual trajectory returns, weighted appropriately for temporal proximity, thereby providing a reliable learning signal for the actor.

The actor represents a stochastic policy, $\pi_\phi(a_t|s_t)$, parameterized by $\phi$, which outputs a distribution over actions conditioned on the current state. It is optimized to select actions that maximize expected returns, as evaluated by the critic, using a policy gradient objective. Specifically, the actor is trained by minimizing the following surrogate loss:

$$J_\pi(\theta) = \mathbb{E}_{s_t \sim \mathcal{D}} \left[ \log \pi_\phi(a_t|s_t) - Q_\theta(s_t, a_t) \right] \quad (12)$$

Eq. 12 promotes higher log-probability for actions that lead to lower critic-assigned Q-values, effectively improving the policy by increasing the expected advantage. The coupling between the actor and critic enables efficient policy learning, where the critic guides the actor toward actions with greater long-term utility in the sequential task allocation setting.

**Complexity Considerations**   Due to the step-wise nature of the sequential assignment process, each environment rollout requires $O(|T|)$ evaluations, where $|T|$ is the number of tasks. However, this also enables finer control over constraint adherence and policy refinement at each step. We illustrate this for a simple 2-agent, 2-task environment in Fig. 8, which generalizes to larger instances through scalable GNN representations.

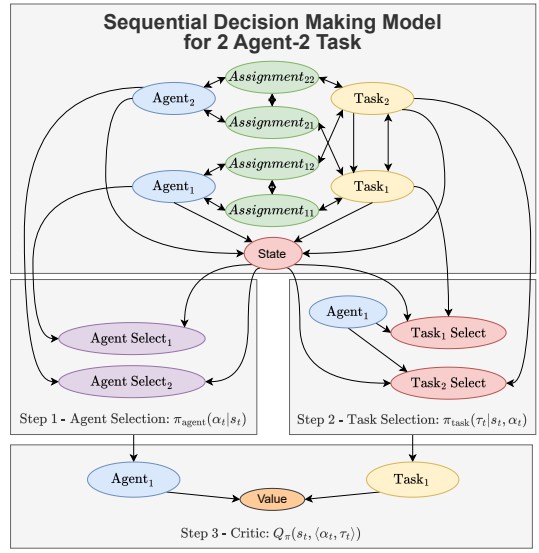

Figure 8: Sequential Decision Making for 2 Agent, 2 Task Problem, utilizing the relational environment representation to (1) select a agent, $\alpha_t$ given state, $s_t$ at step $t$, (2) select a task, $\tau_t$ given state, $s_t$, and agent, $\alpha_t$ at step $t$ (3) evaluate the assignment of task, $\tau_t$, to agent, $\alpha_t$, for training using the Critic output, $Q_t$, at step $t$

.

## Appendix D    Training and Testing Algorithms

### D.1    Simultaneous Model Training

We jointly train a composite task allocation and scheduling policy $\pi_\phi$, consisting of a task allocation policy $\pi_{TA}$ and a scheduling policy $\pi_{Sch}$, using the REINFORCE algorithm [89] with log proba-bilities. The inputs are the problem set, $P$, total training steps, $N$, batch size, $K$ (Line 1), aiming to learn the simultaneous policy, $\pi_\phi$ (Line 2). The parameters of the policy are initialized (Line 3), and for each training iteration, $i \in [1, N]$, a problem, $p$, is sampled from problem set, $P$, and used to initialize the environment (Line 4). Empty sets are initialized for the schedule, $S$, and reward trajectory, $R$, (Line 5), and the number of tasks, $|T_p|$, is determined from the problem instance (Line 6). The initial environment state, $s_0^i$ is observed (Line 7), and both task allocation logits, $o_{TA}$, and scheduling logits, $o_{Sch}$, are generated using the respective policies conditioned on initial state, $s_0^i$, (Lines 8-9). For each task timestep, $t \in [0, |T_p|)$, a task, $\tau_t$, is sampled from the softmax distribution over scheduling logits, $o_{Sch}$, (Line 11), and an agent, $\alpha_t$, is sampled from the softmax distribution of the task allocation logits, $o_{TA}(\tau_t)$, for the specific task (Line 12). The action, $a_t = \langle \alpha_t, \tau_t \rangle$, is taken, and the reward, $R_t^i$, is observed (Line 13). The scheduled task is removed from set of scheduling logits, $o_{Sch}$, (Line 14), and the action and reward are appended to the schedule and reward lists (Line 15). After completing the rollout for all tasks, the policy parameters, $\phi$, are updated using the REINFORCE policy gradient, scaled by a learning rate, $\lambda_\pi$, (Line 16). The training loop continues for a global step, $N$, iterations to optimize both task allocation and scheduling simultaneously.

---

**Algorithm 1** Simultaneous Policy training with REINFORCE

---

1: **Input:** problem-set $P$, global steps $N$, batch-size $K$
2: **Output:** Scheduler policy $\pi_\phi$, consisting of task allocation, $\pi_{TA}$, and scheduling policy, $\pi_{Sch}$
3: Initialize parameters for scheduler policy $\pi_\phi$
4: **for** $i = 1$ **to** $N$ **do**
5:     Initialize the environment with the problem $p \in P$
6:     Initialize episode schedule, $S = \{\}$, rewards, $R = \{\}$
7:     Get number of tasks to determine schedule size $|T_p| \leftarrow p$
8:     Observe state $s_0^i$
9:     Get Task Allocation output, $o_{TA} = \pi_{TA}(\cdot | s_0)$
10:     Sample the Agent assignment for each Task, $A \sim softmax(o_{TA})$
11:     Get Scheduling output conditioned on allocation, $o_{Sch} = \pi_{Sch}(\cdot \mid s_0, A)$
12:     **for** $t \in [0, |T_p|)$ **do**
13:         Sample task, $\tau_t \sim softmax(o_{Sch})$
14:         Get sampled agent for the task $\tau_t$, $\alpha_t = A(\tau_t)$.
15:         Set action, $a_t = \langle \alpha_t, \tau_t \rangle$, observing reward , $R_t^i$.
16:         Remove assigned task from schedule $o_{Sch} = o_{Sch} \setminus \tau_t$
17:         Update schedule, $S = S \cup a_t$, and reward, $R = R \cup R_t$
18:     **end for**
19:     Update actor parameters $\phi = \phi - \lambda_\pi \widehat{\nabla}_\phi J_\pi(\phi)$
20: **end for**

---

### D.2    Sequential Model Training

We train the Sequential Model using the Soft Actor-Critic Algorithm presented in Algorithm 2, on a given set of scheduling problems, $P$. The inputs are the problem set, $P$, total training steps, $N$, batch size, $K$, and actor start training time, $N_A$ (Line 1), aiming to learn the sequential policy, $\pi_\phi$ (Line 2). For each problem instance, $p \in P$, the environment is initialized accordingly (Line 4), and the episode-level schedule record, $S$, is initialized as an empty set (Line 5). The initial state, $s_0^i$, is observed from the environment based on problem, $p$, (Line 6). The inner loop iterates over decision steps, $t \in [0, |T_p|)$, where at each step an action $a_t^p = \langle \alpha_t, \tau_t \rangle$, is the agent, $\alpha_t$ and task, $\tau_t$, pairing, selected by taking the `argmax` over the stochastic policy distribution, $\pi_\phi(s_t^p)$, (Line 8). This action is executed in the environment, yielding the next state, $s_{t+1}^p$, immediate reward, $R_t^p$. The tuple $\{s_t^p, a_t^p, r_t^p, f_t^p, s_{t+1}^p\}$ is appended to the schedule, $S$, (Line 9). After all decisions have been made, the final episode reward, $R_{|T_p|}^p$ is aggregated into the overall reward set $R$ (Line 11). The

training loop continues for a global step, $N$, iterations to optimize the sequential task allocation and scheduling algorithm.

---

**Algorithm 2** Sequential Policy training with Soft Actor-Critic (SAC)

---

1: **Input:** problem-set $P$, global steps $N$, batch-size $K$, actor start training time $N_A$
2: **Output:** Scheduler policy $\pi_\phi$, consisting of agent-select and task-select policy
3: Initialize parameters for scheduler policy $\pi_\phi$, critic $Q_{\theta_1}, Q_{\theta_2}$, target-critics $Q_{\theta_1'}, Q_{\theta_2'}$ and replay-buffer $D$
4: **for** $i = 1$ to $N$ **do**
5:      Initialize the environment with the problem $p \in P$
6:      Initialize episode schedule $S = \{\}$
7:      Get number of tasks to determine schedule size $|T_p| \leftarrow p$
8:      Observe state $s_0^i$
9:      **for** $t \in [0, |T_p|)$ **do**
10:         Sample and execute $a_t^i \sim \pi_\phi(s_t^i)$, observing state, $s_{t+1}^i$ reward , $r_t^i$, feasibility, $f_t^i$.
11:         $S \leftarrow S \cup \{s_t^i, a_t^i, r_t^i, s_{j+1}^i\}$
12:      **end for**
13:      $D = D \cup \{s_t^i, a_t^i, r_{T_p}^i, s_{t+1}^i\} \; \forall t \in 1, ..., T_p$
14:      **if** gradient update step **then**
15:         Update critic parameters $\theta_i = \theta_i - \lambda_{\theta_i} \widehat{\nabla}_{\theta_i} J_Q(\theta_i)$ where $i \in \{1, 2\}$
16:         **if** $i > N_A$ **then**
17:            Update actor parameters $\phi = \phi - \lambda_\pi \widehat{\nabla}_\phi J_\pi(\phi)$
18:         **end if**
19:      **end if**
20: **end for**

---

### D.3 Testing and Evaluation

We evaluate our scheduling policies, which include heuristic, metaheuristic, and learned schedulers, based on Algorithm 3. The algorithm receives as input the learned policy, $\pi_\phi$, and the scheduling problem set, $P$, (Line 1), and produces the final reward and feasibility metrics across all instances (Line 2). Initialize the reward, $R$, and feasibility, $F$, records as empty sets (Line 3) For each problem, $p \in P$, the scheduling environment is initialized (Line 5), the initial state, $s_0^i$, is derived from the problem instance, $p$, (Line 6). If the policy is simultaneous (Line 7), the complete task allocation and schedule, $S$, is generated using the policy, $\pi_\phi(s_0^p)$ based on initial observation, $s_0^p$ (Line 8). If the policy is sequential (Line 9), an empty schedule trajectory, $S$, is created to store per-step information (Line 10). Over a sequence of scheduling decisions, $|T_p|$, (Line 11), the policy, $\pi_\phi$, selects an action, $a_t^p = \langle \alpha_t, \tau_t \rangle$, via a greedy selection, i.e., the argmax over all possible agent-task pairs based on the policy output for the current state, $s_t^p$. Upon executing the action, the next state, $s_{t+1}^p$, and corresponding reward, $r_t^p$, are observed (Line 12). The action is appended to the schedule, $S$ (Line 13). After completing all decision steps, the final reward, $R_{|T^p|}$, and the number of feasible allocations, $F^p$, within $S$ is counted (Line 16), and added to the record. This loop repeats for all problems in $P$, allowing statistical evaluation of the scheduling policy's effectiveness and constraint adherence.

## Appendix E  Datasets and Experiments

This section outlines the synthetic data generation process (Appendix E.1), the training and evaluation details (Appendix E.2) used to benchmark our proposed methods. Detailed hyperparameter configurations, environment settings, and platform specifications are provided to support replicability and comparative analysis.

### E.1 Data Generation

We define the simulation environment using configurable parameters: map dimensions $(w, h)$, number of agents $|A|$, and number of tasks $|T|$. Agents and tasks are placed uniformly at random within the map area.

**Algorithm 3** Scheduling Policy Evaluation Algorithm

1: **Input:** Scheduling Policy $\pi_\phi$, problem set $P$
2: **Output:** Reward $R$, Number of Feasible Task Assignments $F$
3: Initialize reward, $R = \{\}$, feasibility cound, $F = \{\}$.
4: **for** $p \in P$ **do**
5:    Initialize the environment with the problem $p \in P$
6:    Observe state $s_0^p \leftarrow p$
7:    **if** $\pi_\phi$ is Simultaneous **then**
8:       Get complete schedule $S \sim \pi_\phi(s_0^p)$
9:    **else**
10:      Initialize episode schedule $S = \{\}$
11:      **for** $t \in [0, |T_p|)$ **do**
12:         Get action and execute $a_t^p \sim \text{argmax}_{(\alpha_t \in A, \tau_t \in T)} \pi_\phi(s_t^p)$, observing state, $s_{t+1}^p$.
13:         $S \leftarrow S \cup \{a_t^p\}$
14:      **end for**
15:    **end if**
16:    Add reward, $R = R \cup \{R_{|T_p|}\}$, and number of feasible assignments, $F = F \cup \{F^p\}$
17: **end for**

|  | Number of Agents | Number of Tasks | Minimum Time Window (%) | Maximum Time Window (%) | Tasks with Wait Time (%) |
|---|---|---|---|---|---|
| **Small** | 10 | 20 | 0 | 10 | 25 |
| **Medium** | 10 | 50 | 0 | 10 | 25 |
| **Large** | 20 | 100 | 0 | 10 | 25 |
| **Extra Large** | 40 | 200 | 0 | 10 | 25 |

Table 1: Dataset Generation representing the scales based on number of agents and number of tasks along with range of constraints being used.

The total deadline $t_{ddl}$ is computed based on the maximum travel time and maximum task execution time, is calculated based on the maximum travel time, $t_{ij}^T$, from agent, $i$, start position to task, $j$, location, travel time between two tasks, $t_{ijk}^T$, from task $j$ to task $k$, by agent, $i$, and maximum execution time, $t_{ij}^E$, of task, $j$, by agent, $i$, as described in Eq. 13 for $|T|$ tasks.

$$t_{ddl} = |T| \left( \max \left( \max_{i \in \mathcal{A}, j \in \mathcal{T}}(t_{ij}^T), \max_{i \in \mathcal{A}, j, k \in \mathcal{T}}(t_{ijk}^T) \right) + \max_{i \in \mathcal{A}, j \in \mathcal{T}}(t_{ij}^E) \right) \tag{13}$$

The time-windows are generated by randomly generating a time-percentage $t_w$ between $t_{w_{min}}$ and $t_{w_{max}}$. The task start window is uniformly sampled from $t_j^S \sim [0, t_{ddl} - t_w * t_{ddl}]$, and task end window is set to $t_j^E = t_j^S + t_w t_{ddl}$.

Task durations are sampled from $\sim U(10, 100)$ for each agent-task pair. The wait-time percentage of tasks that have wait-time constraints and the duration for the wait-time constraints are sampled from $\sim U(10, 100)$.

We generate four scales based on the number of agents and number of tasks, as shown in Table 1, using the MILP Formulation described in Appendix A to validate the existence of fully feasible solutions. Due to the complexity of the Large and Extra-Large scale problems, we generate 2 and 4 medium-scale problems in the same map, overlaying them to create the Large and Extra-Large scale problems, respectively. The time window range, $t_{w_{min}}, t_{w_{max}}$, and the percentage of tasks with wait time constraints are also shown in Table 1.

### E.2 Training and Evaluation Details

We train all models exclusively using data from the small-scale setting and evaluate their performance across four distinct scales: small, medium, large, and x-large. Experiments for small to large scales were conducted on a Mac Studio equipped with an Apple M1 chip and 32 GB of RAM. Due to

increased memory demands during optimization, the extra-large scale evaluations were performed on a high-performance server featuring an AMD EPYC 7452 processor running Ubuntu 20.04.6.

All models employ a four-layer architecture, as informed by prior empirical analysis and established methods in the literature [61]. For attention mechanisms, we adopt a multi-head configuration with eight heads for sequential models [24, 61], while simultaneous models are configured with a single head to reduce computational overhead. Model training uses a learning rate of 0.001 and an entropy coefficient of 0.01. The Critic network is pre-trained for 5,000 steps, after which Actor-Critic training proceeds for 25,000 steps, where each step represents a single task-agent assignment decision.

## Appendix F    Baselines

We compare TARGETNET using Edge Enhanced HGT with Residuals against Heuristic and Meta-heuristic schedulers, and compare the model performance against existing Graph-based Learning Models, that leverage the Task Allocation Graph Representation Model presented in Fig. 7.

**Exact Solvers**

- **Mixed-Integer Linear Program (MILP)**: Exact solver which was also utilized for problem validation described in Eq. 7. The full implementation details can be found in Appendix A. If the MILP Solver is not able to find an optimal solution within a given time limit (12 hours), the system returns the best partial schedule. We utilize Gurobi Solver [48] to solve the Constraint Satisfaction Problem defined in Eq. 7.

**Heuristic Solvers**

- **Earliest Deadline First (EDF)**: A heuristic method for greedy agent-task assignment that prioritizes the earliest deadline, without checking wait-time constraints on tasks  [91]. The heuristic dynamically picks the agents as they become available.

- **Constraint-Aware EDF (CA-EDF)**: A variant of **EDF** that accounts for task dependencies, only selecting tasks with all prerequisites completed [26, 51]. The optimizer returns the most optimal schedule if time-limit is reached.

**Metaheuristics**

- **Genetic Algorithm**: A metaheuristic that improves a population of schedules over several generations by applying mutations and selecting the best schedules based on an initial seed [30, 29]. We evaluate the performance of **Gen-Random** (starting from a random schedule) and **Gen-EDF** (starting from an **EDF**-based schedule). Given the high time complexity of the Genetic Algorithm, we limit the evaluation to the best-performing schedules after 1 and 3 generations of mutations based on computation time to be comparable to the optimal solver, with a population size of 100 and survival rate of 10% for the next generation.

**Learning methods - Sequential Decision Making**   We compare our model against sequential decision models as described in prior works [61], comparing our method to Heterogenous Graph Attention Networks and Heterogenous Graph Transformer-based models. The implementation and training details can be found in Appendix C.

- **Heterogeneous Graph Attention Network (HetGAT)**: A method for sequential agent-task assignment from Wang et al. [61]. The Graph Model used is modified to account for the travel time and task assignment as per Fig. 8.

- **HetGAT with Residual Connections (Res-HetGAT)**: The HetGAT model enhanced with residual connections from the input layer to each subsequent layer as presented in Zhang and Meng [88].

- **Sequential Heterogenous Graph Transformer (SeqTARGETNET\ER)**: Heterogeneous Graph Transformer (HGT) model presented in  Hu et al. [40] without Residuals, applied to sequential agent-task allocation based on Wang et al. [61].

- **Sequential HGT with Edge Attention (Seq-TARGETNET\R)**: A modified version of HGT that incorporates edge attention described in Section 3.4.
- **Sequential TARGETNET (Seq-TARGETNET)**: An HGT model with edge attention and residual connections to the input layer as described in Section 3.4.

**Learning methods: Simultaneous Decision Making**

- **Simultaneous Heterogenous Graph Transformer (TARGETNET\RE)**: Heterogeneous Graph Transformer model presented in Hu et al. [40] without Residuals, applied to simultaneous agent-task allocation and scheduling as per Fig. 2a.
- **Simultaneous HGT with Edge Attention (TARGETNET\R)**: A modified version of HGT that incorporates edge attention as described in Section 3.4.
- **HGT-Edge with Residual Connections (TARGETNET) (Ours)**: The Simultaneous Task Allocation and Scheduling model described in Section 3.5 using the Res-HGT-Edge model with edge attention and residuals as described in Section 3.4.

## Appendix G    Additional Experiments and Results

This appendix provides further insights into the performance and robustness of the proposed models across a range of scenarios and experimental conditions. The complete performance metrics across model variants, as presented in Figures 3, 4, and 6, are summarized in Table 2. We present detailed tabular results complementing the figures shown in the main paper, and expand on two key aspects of evaluation: sensitivity to constraint variations (Section G.1) and training stability (Section G.2).

### G.1    Sensitivity Analysis

Fig. 9 and 10 evaluate the sensitivity of model performance under varying task scheduling constraints. Each dataset in this analysis varies a single constraint type. We vary the time window (tight vs. relaxed), wait-time between tasks (low vs. high), or agent speed (slow vs. fast).

We observe that both sequential and simultaneous graph-based approaches, particularly those using TARGETNET architectures, maintain strong performance in terms of optimality rate and solution feasibility. Notably, these models adapt effectively to different constraint regimes without requiring fine-tuning. This indicates that the models learn meaningful representations of the constraint space, enabling generalization to a wide range of practical scheduling tasks within the bounds of the training distribution.

Fig. 9 and 10 evaluate the sensitivity of model performance under varying task scheduling constraints. Each dataset in this analysis varies a single constraint type. We vary the time window (tight vs. relaxed), wait-time between tasks (low vs. high), or agent speed (slow vs. fast).

We observe that both sequential and simultaneous graph-based approaches, particularly those using TARGETNET architectures, maintain strong performance in terms of optimality rate and solution feasibility. Notably, these models adapt effectively to different constraint regimes without requiring fine-tuning. This indicates that the models learn meaningful representations of the constraint space, enabling generalization to a wide range of practical scheduling tasks within the bounds of the training distribution.

Table 2:

| Scale | Data Type | Exact | Heuristic | | Metaheuristics | | | | | | Sequential | | | | Simultaneous | | | |
|---|---|---|---|---|---|---|---|---|---|---|---|---|---|---|---|---|---|---|
| | | MILP | EDF | CA-EDF | Gen-Random 1 | Gen-EDF 1 | Gen-Random 3 | Gen-EDF 3 | HetGAT | Res-HetGAT | Seq-TARGETNET \ RE | Seq-TARGETNET \ E | Seq-TARGETNET \ R | Seq-TARGETNET | TARGETNET \ RE | TARGETNET \ E | TARGETNET \ R | TARGETNET (Ours) |
| Small | Reward ↑ | 20.08 (0.05) | 13.76 (2.18) | 16.00 (2.18) | 12.83 (2.03) | 15.79 (1.68) | 15.52 (1.70) | 17.44 (1.31) | 14.51 (1.77) | 13.94 (2.02) | 2.10 (0.91) | 7.75 (1.10) | 2.10 (0.91) | 3.26 (1.17) | 2.13 (0.98) | 19.74 (0.57) | 2.10 (0.91) | 19.72 (0.57) |
| | Feasible (%) ↑ | 100.00 (0.00) | 68.40 (10.93) | 79.45 (11.13) | 63.22 (10.38) | 78.42 (8.46) | 76.88 (8.65) | 86.67 (6.58) | 72.00 (8.93) | 68.88 (10.41) | 5.53 (4.57) | 34.38 (5.89) | 5.53 (4.57) | 13.50 (6.63) | 5.65 (4.88) | 98.28 (2.85) | 5.53 (4.57) | 98.17 (2.88) |
| | Optimality Rate (%) ↑ | 100.00 (0.00) | 68.53 (10.86) | 79.66 (10.84) | 63.87 (10.11) | 78.63 (8.35) | 77.25 (8.48) | 86.83 (6.54) | 72.23 (8.84) | 69.41 (10.08) | 10.48 (4.54) | 38.59 (5.45) | 10.48 (4.54) | 16.23 (5.81) | 10.61 (4.86) | 98.31 (2.82) | 10.48 (4.54) | 98.21 (2.86) |
| | Computation Time (s) ↓ | 1.27 (0.28) | 0.07 (0.03) | 0.07 (0.03) | 6.82 (0.02) | 6.95 (0.02) | 19.20 (0.14) | 19.18 (0.09) | 3.12 (0.74) | 2.66 (0.08) | 3.79 (0.74) | 2.64 (0.06) | 6.33 (0.73) | 5.73 (0.17) | 0.08 (0.00) | 0.09 (0.03) | 0.11 (0.02) | 0.10 (0.01) |
| Medium | Reward ↑ | 50.06 (0.03) | 15.79 (3.30) | 19.44 (3.07) | 17.70 (2.85) | 17.75 (2.91) | 21.27 (2.62) | 20.51 (2.92) | 24.50 (2.69) | 18.42 (3.42) | 2.38 (1.10) | 8.09 (1.45) | 2.38 (1.10) | 3.46 (1.09) | 2.38 (1.10) | 49.00 (1.20) | 2.38 (1.10) | 48.96 (1.15) |
| | Feasible (%) ↑ | 100.00 (0.00) | 29.59 (6.60) | 36.88 (6.14) | 34.39 (5.86) | 34.56 (5.76) | 41.85 (5.30) | 40.45 (5.87) | 47.37 (5.52) | 34.97 (6.95) | 2.76 (2.20) | 14.18 (2.91) | 2.76 (2.20) | 5.56 (2.50) | 2.76 (2.20) | 97.88 (2.41) | 2.76 (2.20) | 97.81 (2.29) |
| | Optimality Rate (%) ↑ | 100.00 (0.00) | 31.55 (6.60) | 38.84 (6.14) | 35.36 (5.70) | 35.46 (5.82) | 42.48 (5.23) | 40.97 (5.84) | 48.95 (5.37) | 36.80 (6.84) | 4.75 (2.19) | 16.16 (2.90) | 4.75 (2.19) | 6.92 (2.18) | 4.75 (2.19) | 97.88 (2.40) | 4.75 (2.19) | 97.81 (2.29) |
| | Computation Time (s) ↓ | 17.21 (8.15) | 1.51 (0.04) | 1.53 (0.10) | 171.37 (10.14) | 160.81 (2.78) | 435.77 (1.39) | 438.93 (0.73) | 20.21 (0.08) | 20.18 (0.11) | 19.19 (0.06) | 15.72 (0.47) | 31.63 (0.14) | 31.81 (0.14) | 0.22 (0.04) | 0.22 (0.03) | 0.26 (0.03) | 0.28 (0.04) |
| Large | Reward ↑ | 98.31 (4.54) | 30.07 (5.74) | 38.43 (4.92) | 31.71 (3.37) | 32.09 (5.43) | 34.71 (3.69) | 34.75 (5.21) | 43.31 (4.35) | 45.82 (4.27) | 2.13 (1.09) | 14.10 (2.41) | 2.13 (1.09) | 3.85 (1.20) | 2.13 (1.09) | 97.58 (3.48) | 2.13 (1.09) | 97.81 (1.76) |
| | Feasible (%) ↑ | 98.27 (4.55) | 29.07 (5.74) | 37.43 (4.92) | 31.47 (3.39) | 31.67 (5.42) | 34.47 (3.75) | 34.43 (5.25) | 42.63 (4.45) | 44.97 (4.41) | 1.13 (1.09) | 13.10 (2.41) | 1.13 (1.09) | 3.40 (1.31) | 1.13 (1.09) | 97.53 (3.48) | 1.13 (1.09) | 97.77 (1.76) |
| | Optimality Rate (%) ↑ | 100.00 (0.00) | 30.66 (6.06) | 39.12 (4.95) | 32.34 (3.81) | 32.72 (5.75) | 35.43 (4.53) | 35.44 (5.61) | 44.19 (5.23) | 46.68 (4.51) | 2.19 (1.21) | 14.37 (2.52) | 2.19 (1.21) | 3.95 (1.32) | 2.19 (1.21) | 99.50 (6.29) | 2.19 (1.21) | 99.70 (4.71) |
| | Computation Time (s) ↓ | 488.10 (305.58) | 23.73 (0.25) | 23.45 (0.16) | 2327.78 (63.89) | 2360.06 (48.83) | 6980.01 (95.51) | 7249.58 (31.27) | 239.62 (3.88) | 323.97 (125.75) | 271.52 (6.10) | 241.74 (2.37) | 505.33 (50.06) | 481.99 (12.59) | 1.05 (0.03) | 1.05 (0.04) | 1.32 (0.06) | 1.30 (0.05) |
| XLarge | Reward ↑ | 54.27 (5.19) | 62.20 (6.05) | 75.10 (7.57) | 56.31 (3.86) | 63.73 (6.07) | 63.56 (3.78) | 65.16 (7.56) | 80.32 (6.61) | 102.35 (5.16) | 2.80 (1.17) | 26.80 (2.52) | 2.80 (1.17) | 4.23 (1.59) | 2.80 (1.17) | 196.43 (1.56) | 2.80 (1.17) | 195.73 (1.61) |
| | Feasible (%) ↑ | 27.05 (2.62) | 30.60 (3.02) | 37.05 (3.78) | 28.05 (1.96) | 31.55 (2.97) | 31.70 (1.86) | 32.45 (3.80) | 39.80 (3.39) | 50.80 (2.54) | 0.90 (0.58) | 12.90 (1.26) | 0.90 (0.58) | 1.80 (0.84) | 0.90 (0.58) | 98.20 (0.78) | 0.90 (0.58) | 97.85 (0.81) |
| | Optimality Rate (%) ↑ | 100.00 (0.00) | 115.00 (9.75) | 139.30 (15.97) | 104.48 (10.38) | 117.83 (9.62) | 118.00 (12.24) | 121.12 (17.95) | 148.77 (13.11) | 190.06 (18.29) | 5.33 (2.62) | 49.72 (6.02) | 5.33 (2.62) | 7.83 (2.78) | 5.33 (2.62) | 365.09 (33.34) | 5.33 (2.62) | 363.73 (32.54) |
| | Computation Time (s) ↓ | 6232.24 (5041.77) | 313.85 (16.74) | 312.78 (16.03) | 27909.77 (43.01) | 29309.63 (2203.89) | 78173.11 (257.43) | 78504.93 (279.11) | 3226.15 (219.93) | 3151.54 (293.03) | 3094.58 (269.81) | 3209.68 (6.76) | 6451.99 (798.16) | 6457.32 (536.18) | 6.29 (0.09) | 6.29 (0.12) | 7.85 (0.18) | 7.96 (0.14) |

Table 2: Mean Scores (Standard Deviations in Parentheses below) and Empirical Time Performance of heuristics, metaheuristics and graph-based learned policies using TARGETNET, trained on small scale and evaluated on different scales, as presented in Fig. 3 and 4 The results show that our models outperform baselines and scales to larger models better than both heuristics and metaheuristics. ↑ and ↓ indicates higher is better and lower is better respectively.

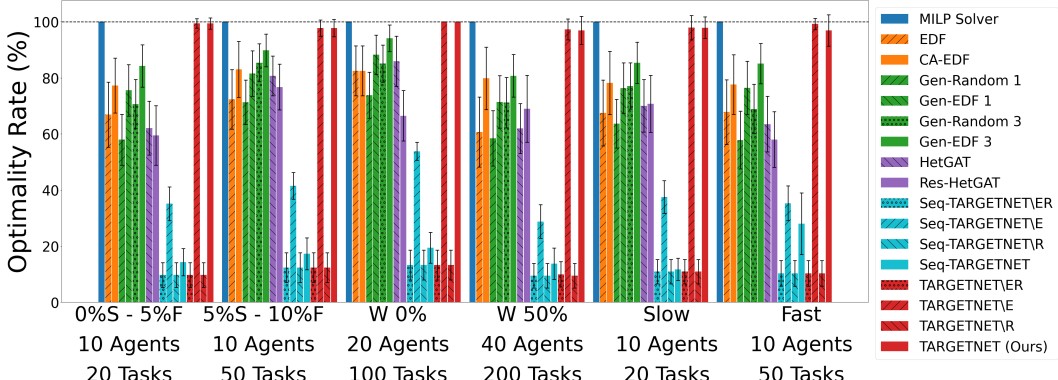

Figure 9: Mean optimality rate of the final reward across various constraint settings. The evaluation reflects both Makespan and the number of feasible task assignments. Results indicate that both sequential and simultaneous graph-based models maintain high performance across different constraint regimes. Higher values indicate better performance.

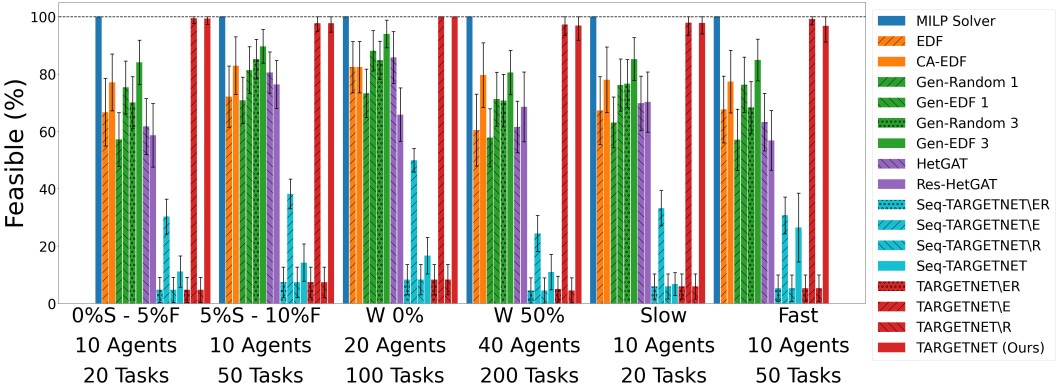

Figure 10: Percentage of feasible solutions achieved under varying constraint settings. The graph-based models generalize well to different constraint ranges, achieving high feasibility across a range of scenarios. Higher is better.

## G.2 Training Stability

While we report the best-performing model from an ensemble of three random seeds (seeds 10, 11, and 12) in Section 5, Table 4 presents a breakdown of how learning-based models perform when initialized with different seeds. This analysis is crucial for understanding the sensitivity of our models to random initialization.

In the small and medium scales, seed 10 for TARGETNET consistently achieves strong performance. However, in the medium to extra-large scales, seed 11 produces the most effective policy, indicating that no single seed consistently dominates across all problem sizes.

The sensitivity analysis was conducted across three random seeds to evaluate the robustness of the learning-based models, as presented in the box plot of Fig. 11. The results show that TARGETNET consistently achieves higher median performance, and lower variance compared to its ablated variants, indicating more stable learning behavior across different training initializations. On average, TARGETNET learns policies that outperform those of the ablation models, demonstrating the effectiveness of jointly leveraging both node and edge features through its graph-based attention mechanism. This stability across constraint ranges and under varying seeds further supports the model's capacity for robust policy generalization.

Using an ensemble of seeds allows us to capture a broader set of learned policies, each potentially discovering different solution strategies. Due to the exploratory nature of policy learning and the randomness in optimization trajectories, models initialized with different seeds may settle on varying performance plateaus. Some seeds may result in policies that get stuck in suboptimal behaviors, while others may discover more efficient task allocation strategies. By evaluating multiple seeds and

Table 3 top grouping: Exact | Heuristic | Metaheuristics | Sequential | Simultaneous

| Scale | Data Type | MILP | EDF | CA-EDF | Gen-Random 1 | Gen-EDF 1 | Gen-Random 3 | Gen-EDF 3 | HetGAT | Res-HetGAT | Seq-TARGETNET\ER | Seq-TARGETNET\E | Seq-TARGETNET\R | Seq-TARGETNET | TARGETNET\ER | TARGETNET\E | TARGETNET\R | TARGETNET (Ours) |
|---|---|---|---|---|---|---|---|---|---|---|---|---|---|---|---|---|---|---|
| **0%S - 5%F** | Reward ↑ | 20.07 (0.05) | 13.42 (2.33) | 15.50 (1.96) | 11.63 (1.82) | 15.16 (1.83) | 14.17 (1.76) | 16.91 (1.51) | 12.46 (1.93) | 11.94 (2.13) | 1.95 (0.89) | 7.05 (1.20) | 1.95 (0.89) | 2.87 (0.97) | 1.95 (0.89) | **19.96 (0.34)** | 1.95 (0.89) | *19.95 (0.41)* |
| | Feasible (%) ↑ | 100.00 (0.00) | 66.70 (11.79) | 77.10 (9.92) | 57.23 (9.38) | 75.38 (9.25) | 70.15 (9.00) | 84.12 (7.72) | 61.72 (9.78) | 58.67 (11.05) | 4.73 (4.45) | 30.30 (6.08) | 4.73 (4.45) | 11.15 (5.47) | 4.73 (4.45) | **99.45 (1.72)** | 4.73 (4.45) | *99.40 (2.03)* |
| | Optimality Rate (%) ↑ | 100.00 (0.00) | 66.89 (11.61) | 77.24 (9.78) | 57.97 (9.08) | 75.55 (9.15) | 70.63 (8.78) | 84.29 (7.55) | 62.06 (9.59) | 59.51 (10.63) | 9.69 (4.43) | 35.12 (5.98) | 9.69 (4.43) | 14.32 (4.85) | 9.69 (4.43) | **99.45 (1.71)** | 9.69 (4.43) | *99.40 (2.03)* |
| **5%S - 10%F** | Reward ↑ | 20.11 (0.04) | 14.54 (2.14) | 16.70 (2.00) | 14.34 (1.60) | 16.40 (1.63) | 17.17 (1.36) | 18.06 (1.17) | 16.23 (1.43) | 15.43 (1.63) | 2.48 (1.07) | 8.35 (1.07) | 2.48 (1.07) | 3.48 (1.15) | 2.48 (1.07) | **19.66 (0.58)** | 2.48 (1.07) | *19.66 (0.61)* |
| | Feasible (%) ↑ | 100.00 (0.00) | 72.17 (10.67) | 82.97 (10.04) | 70.85 (8.13) | 81.42 (8.21) | 85.20 (6.89) | 89.70 (5.87) | 80.60 (7.15) | 76.40 (8.37) | 7.38 (5.34) | 38.22 (5.19) | 7.38 (5.34) | 14.25 (6.48) | 7.38 (5.34) | **97.75 (2.90)** | 7.38 (5.34) | *97.75 (3.07)* |
| | Optimality Rate (%) ↑ | 100.00 (0.00) | 72.32 (10.61) | 83.07 (9.97) | 71.29 (7.94) | 81.57 (8.10) | 85.40 (6.75) | 89.82 (5.81) | 80.73 (7.10) | 76.75 (8.13) | 12.31 (5.30) | 41.54 (4.75) | 12.31 (5.30) | 17.30 (5.70) | 12.31 (5.30) | **97.76 (2.89)** | 12.31 (5.30) | *97.76 (3.05)* |
| **W 0%** | Reward ↑ | 20.08 (0.04) | 16.57 (1.79) | 16.57 (1.79) | 14.82 (1.64) | 17.72 (1.40) | 17.09 (1.32) | 18.91 (0.95) | 17.26 (1.80) | 13.36 (1.80) | 2.67 (1.06) | 10.81 (0.65) | 2.67 (1.06) | 3.90 (1.11) | 2.67 (1.06) | **20.08 (0.04)** | 2.67 (1.06) | *20.08 (0.04)* |
| | Feasible (%) ↑ | 100.00 (0.00) | 82.45 (8.96) | 82.45 (8.96) | 73.32 (8.47) | 88.15 (7.04) | 84.82 (6.72) | 94.08 (4.82) | 85.82 (9.07) | 65.88 (9.31) | 8.32 (5.30) | 49.92 (4.08) | 8.32 (5.30) | 16.65 (6.37) | 8.32 (5.30) | **100.00 (0.00)** | 8.32 (5.30) | *100.00 (0.00)* |
| | Optimality Rate (%) ↑ | 100.00 (0.00) | 82.54 (8.91) | 82.54 (8.91) | 73.81 (8.20) | 88.24 (6.97) | 85.10 (6.59) | 94.15 (4.76) | 85.95 (8.97) | 66.51 (8.96) | 13.27 (5.28) | 53.83 (3.23) | 13.27 (5.28) | 19.44 (5.51) | 13.27 (5.28) | **100.00 (0.00)** | 13.27 (5.28) | *100.00 (0.01)* |
| **W 50%** | Reward ↑ | 20.09 (0.05) | 12.19 (2.51) | 12.19 (2.24) | 11.75 (1.99) | 14.35 (1.89) | 14.30 (1.81) | 16.22 (1.53) | 12.45 (1.78) | 13.86 (2.39) | 1.91 (0.88) | 5.78 (1.20) | 1.91 (0.88) | 2.76 (1.13) | 2.00 (0.91) | **19.55 (0.75)** | 1.91 (0.88) | *19.48 (1.01)* |
| | Feasible (%) ↑ | 100.00 (0.00) | 60.50 (12.53) | 79.67 (11.31) | 57.82 (10.17) | 71.28 (9.45) | 70.75 (9.18) | 80.58 (7.70) | 61.58 (8.98) | 68.60 (12.16) | 4.53 (4.40) | 24.38 (6.26) | 4.53 (4.40) | 10.95 (6.17) | 5.00 (4.53) | **97.30 (3.74)** | 4.53 (4.40) | *96.95 (5.07)* |
| | Optimality Rate (%) ↑ | 100.00 (0.00) | 60.69 (12.47) | 79.84 (11.12) | 58.47 (9.89) | 71.45 (9.39) | 71.19 (9.02) | 80.74 (7.61) | 61.95 (8.86) | 69.00 (11.91) | 9.48 (4.38) | 28.76 (5.99) | 9.48 (4.38) | 13.75 (5.61) | 9.95 (4.50) | **97.31 (3.72)** | 9.48 (4.38) | *96.97 (5.04)* |
| **Slow** | Reward ↑ | 20.09 (0.05) | 13.56 (2.36) | 15.71 (2.27) | 12.79 (1.74) | 15.34 (1.81) | 15.49 (1.66) | 17.15 (1.49) | 14.07 (1.89) | 14.21 (2.04) | 2.19 (0.87) | 7.54 (1.18) | 2.19 (0.87) | 2.35 (0.80) | 2.19 (0.87) | **19.67 (0.87)** | 2.19 (0.87) | *19.67 (0.77)* |
| | Feasible (%) ↑ | 100.00 (0.00) | 67.30 (11.84) | 78.03 (11.48) | 63.18 (8.85) | 76.20 (9.09) | 76.70 (8.40) | 85.25 (7.48) | 69.83 (9.48) | 70.28 (10.53) | 5.95 (4.37) | 33.25 (6.14) | 5.95 (4.37) | 6.77 (4.00) | 5.95 (4.37) | **97.92 (4.37)** | 5.95 (4.37) | *97.90 (3.88)* |
| | Optimality Rate (%) ↑ | 100.00 (0.00) | 67.48 (11.73) | 78.21 (11.29) | 63.69 (8.67) | 76.38 (8.99) | 77.09 (8.24) | 85.38 (7.38) | 70.05 (9.38) | 70.76 (10.15) | 10.90 (4.35) | 37.51 (5.84) | 10.90 (4.35) | 11.72 (3.98) | 10.90 (4.35) | **97.94 (4.34)** | 10.90 (4.35) | *97.91 (3.87)* |
| **Fast** | Reward ↑ | 20.08 (0.05) | 13.63 (2.31) | 15.59 (2.13) | 11.62 (2.07) | 15.36 (1.91) | 13.84 (1.77) | 17.09 (1.44) | 12.76 (1.99) | 11.65 (1.99) | 2.07 (0.92) | 7.09 (1.25) | 2.07 (0.92) | 5.62 (2.21) | 2.07 (0.92) | **19.93 (0.40)** | 2.07 (0.92) | *19.46 (1.13)* |
| | Feasible (%) ↑ | 100.00 (0.00) | 67.67 (11.62) | 77.42 (10.89) | 57.10 (10.70) | 76.30 (9.62) | 68.40 (9.00) | 84.97 (7.25) | 63.28 (10.03) | 56.88 (10.41) | 5.35 (4.60) | 30.73 (6.39) | 5.35 (4.60) | 26.52 (12.00) | 5.35 (4.60) | **99.22 (2.01)** | 5.35 (4.60) | *96.88 (5.62)* |
| | Optimality Rate (%) ↑ | 100.00 (0.00) | 67.84 (11.51) | 77.63 (10.61) | 57.85 (10.30) | 76.46 (9.52) | 68.92 (8.82) | 85.11 (7.16) | 63.53 (9.92) | 58.00 (9.92) | 10.31 (4.58) | 35.28 (6.20) | 10.31 (4.58) | 27.98 (11.03) | 10.31 (4.58) | **99.23 (1.99)** | 10.31 (4.58) | *96.90 (5.58)* |

Table 3: Sensitivity Analysis of models trained on Small scale, tested with different constraint ranges and travel time, with Mean Scores (Standard Deviations in Parenthesis below). Our sequential model outperform against heuristics and get comparable performance to metaheuristics. We show the performance in small scale with 0 to 5% time window vs 5 to 10% time window, 0% wait-time vs 50% wait-time, and fast vs slow agents (10 to 20% vs 90 to 100% maximum speed. ↑ and ↓ indicates higher is better and lower is better respectively.

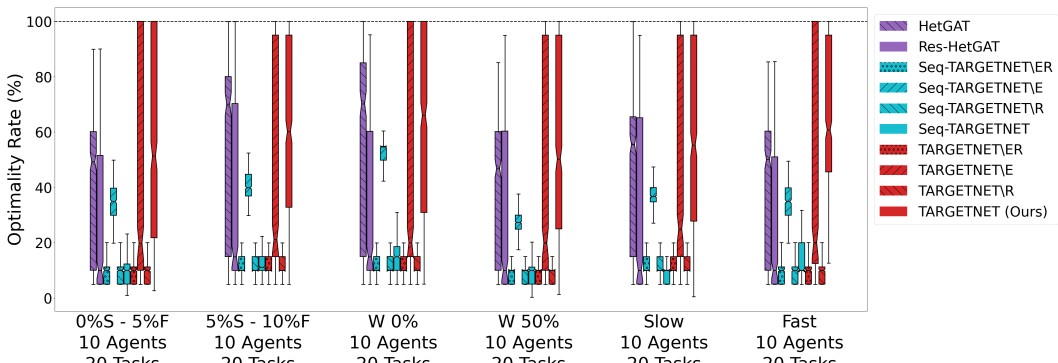

Figure 11: Optimality rate of solutions achieved under varying constraint settings across 3 seeds for the learning based models. The graph-based models generalize well to different constraint ranges, achieving higher performance across a range of scenarios. TARGETNET consistently maintains high optimality rates across all seeds and constraint settings, demonstrating both robustness and reliable generalization compared to ablated models. Higher is better.

selecting the best among them, we reduce the risk of reporting outcomes based on an unrepresentative or poorly initialized run. This ensemble-based approach ensures that the final reported performance more accurately reflects the model's potential and robustness across runs.

Table 4 — Performance across different seeds.

| Scale | Data Type | MILP Solver | HetGAT-10 | HetGAT-11 | HetGAT-12 | Res-HetGAT-10 | Res-HetGAT-11 | Res-HetGAT-12 | Seq-TARGETNET\ER-10 | Seq-TARGETNET\ER-11 | Seq-TARGETNET\ER-12 | Seq-TARGETNET\R-10 (ER) | Seq-TARGETNET\R-11 (ER) | Seq-TARGETNET\R-12 (ER) | Seq-TARGETNET\R-10 | Seq-TARGETNET\R-11 | Seq-TARGETNET\R-12 | Seq-TARGETNET-10 | Seq-TARGETNET-11 | Seq-TARGETNET-12 | TARGETNET\ER-10 | TARGETNET\ER-11 | TARGETNET\ER-12 | TARGETNET\R-10 (E) | TARGETNET\R-11 (E) | TARGETNET\R-12 (E) | TARGETNET\R-10 | TARGETNET\R-11 | TARGETNET\R-12 | TARGETNET (Ours)-10 | TARGETNET (Ours)-11 | TARGETNET (Ours)-12 |
|---|---|---|---|---|---|---|---|---|---|---|---|---|---|---|---|---|---|---|---|---|---|---|---|---|---|---|---|---|---|---|---|---|
| **Small** | Reward ↑ | 20.08 (0.05) | 12.38 (2.47) | 14.09 (1.78) | 2.10 (0.92) | 1.72 (0.76) | 1.77 (0.77) | 13.94 (2.02) | 2.10 (0.91) | 2.10 (0.91) | 2.10 (0.91) | 7.51 (1.20) | 7.36 (1.14) | 7.44 (1.19) | 2.10 (0.91) | 2.10 (0.91) | 2.10 (0.91) | 3.09 (1.32) | 1.74 (0.77) | 2.10 (0.89) | 2.10 (0.91) | 2.10 (0.91) | 2.13 (0.98) | 19.74 (0.57) | 3.39 (1.65) | 3.06 (1.14) | 2.10 (0.91) | 2.10 (0.91) | 2.10 (0.91) | 19.72 (0.57) | 11.20 (2.40) | 4.36 (1.86) |
|  | Feasible (%) ↑ | 100.00 (0.00) | 61.08 (12.57) | 69.95 (8.99) | 5.53 (4.59) | 3.60 (3.81) | 3.85 (3.86) | 68.88 (10.41) | 5.53 (4.57) | 5.53 (4.57) | 5.53 (4.57) | 33.20 (6.42) | 31.77 (5.70) | 32.85 (6.15) | 5.53 (4.57) | 5.53 (4.57) | 5.53 (4.57) | 13.10 (7.01) | 3.70 (3.85) | 5.50 (4.47) | 5.53 (4.57) | 5.53 (4.57) | 5.65 (4.88) | 98.28 (2.85) | 12.15 (8.51) | 10.28 (5.71) | 5.53 (4.57) | 5.53 (4.57) | 5.53 (4.57) | 98.17 (2.88) | 55.10 (12.17) | 18.47 (9.65) |
|  | Optimality Rate (%) ↑ | 100.00 (0.00) | 61.65 (12.33) | 70.14 (8.90) | 10.48 (4.57) | 8.57 (3.80) | 8.82 (3.85) | 69.41 (10.08) | 10.48 (4.54) | 10.48 (4.54) | 10.48 (4.54) | 37.38 (5.97) | 36.62 (5.66) | 37.07 (5.91) | 10.48 (4.54) | 10.48 (4.54) | 10.48 (4.54) | 15.37 (6.55) | 8.67 (3.84) | 10.46 (4.45) | 10.48 (4.54) | 10.48 (4.54) | 10.61 (4.86) | 98.31 (2.82) | 16.89 (8.23) | 15.21 (5.68) | 10.48 (4.54) | 10.48 (4.54) | 10.48 (4.54) | 98.21 (2.86) | 55.78 (11.94) | 21.74 (9.24) |
| **Medium** | Reward ↑ | 50.06 (0.03) | 13.65 (2.23) | 24.50 (2.69) | 2.38 (1.10) | 2.27 (0.90) | 2.29 (0.92) | 18.42 (3.42) | 2.38 (1.10) | 2.38 (1.10) | 2.38 (1.10) | 7.95 (1.41) | 7.59 (1.39) | 7.75 (1.47) | 2.38 (1.10) | 2.38 (1.10) | 2.38 (1.10) | 3.19 (1.24) | 2.25 (0.90) | 2.35 (1.10) | 2.38 (1.10) | 2.38 (1.10) | 2.38 (1.10) | 49.00 (1.20) | 4.35 (2.07) | 3.87 (1.49) | 2.38 (1.10) | 2.38 (1.10) | 2.38 (1.10) | 48.96 (1.15) | 15.05 (2.88) | 2.88 (1.27) |
|  | Feasible (%) ↑ | 100.00 (0.00) | 26.59 (4.53) | 47.37 (5.52) | 2.75 (2.19) | 2.54 (1.80) | 2.58 (1.84) | 34.97 (6.95) | 2.76 (2.20) | 2.76 (2.20) | 2.76 (2.20) | 13.90 (2.83) | 13.19 (2.78) | 13.51 (2.94) | 2.76 (2.20) | 2.76 (2.20) | 2.76 (2.20) | 5.28 (2.68) | 2.49 (1.81) | 2.71 (2.20) | 2.76 (2.20) | 2.76 (2.20) | 2.76 (2.20) | 97.88 (2.41) | 6.81 (4.22) | 5.74 (2.99) | 2.76 (2.20) | 2.76 (2.20) | 2.76 (2.20) | 97.81 (2.29) | 29.19 (5.96) | 3.76 (2.55) |
|  | Optimality Rate (%) ↑ | 100.00 (0.00) | 27.28 (4.45) | 48.95 (5.37) | 4.74 (2.19) | 4.53 (1.79) | 4.57 (1.84) | 36.80 (6.84) | 4.75 (2.19) | 4.75 (2.19) | 4.75 (2.19) | 15.88 (2.82) | 15.17 (2.78) | 15.49 (2.94) | 4.75 (2.19) | 4.75 (2.19) | 4.75 (2.19) | 6.36 (2.48) | 4.48 (1.80) | 4.70 (2.20) | 4.75 (2.19) | 4.75 (2.19) | 4.75 (2.19) | 97.88 (2.40) | 8.69 (4.14) | 7.73 (2.99) | 4.75 (2.19) | 4.75 (2.19) | 4.75 (2.19) | 97.81 (2.29) | 30.07 (5.75) | 5.775 (2.55) |
| **Large** | Reward ↑ | 98.31 (4.54) | 28.12 (3.37) | 43.31 (4.35) | 2.17 (1.07) | 1.93 (0.85) | 2.07 (0.81) | 45.82 (4.27) | 2.13 (1.09) | 2.13 (1.09) | 2.13 (1.09) | 13.77 (2.39) | 13.23 (2.46) | 13.57 (2.32) | 2.13 (1.09) | 2.13 (1.09) | 2.13 (1.09) | 3.69 (1.41) | 2.07 (0.77) | 2.13 (1.09) | 2.13 (1.09) | 2.13 (1.09) | 2.13 (1.09) | 97.58 (3.48) | 4.37 (2.26) | 3.77 (2.04) | 2.13 (1.09) | 2.13 (1.09) | 2.13 (1.09) | 97.81 (1.76) | 33.46 (4.10) | 2.43 (0.80) |
|  | Feasible (%) ↑ | 98.27 (4.55) | 27.73 (3.38) | 42.63 (4.45) | 1.17 (1.07) | 0.93 (0.85) | 1.07 (0.81) | 44.97 (4.41) | 1.13 (1.09) | 1.13 (1.09) | 1.13 (1.09) | 12.77 (2.39) | 12.23 (2.46) | 12.57 (2.32) | 1.13 (1.09) | 1.13 (1.09) | 1.13 (1.09) | 3.33 (1.40) | 1.07 (0.77) | 1.13 (1.09) | 1.13 (1.09) | 1.13 (1.09) | 1.13 (1.09) | 97.53 (3.48) | 3.40 (2.29) | 2.77 (2.04) | 1.13 (1.09) | 1.13 (1.09) | 1.13 (1.09) | 97.77 (1.76) | 33.20 (4.09) | 1.43 (0.80) |
|  | Optimality Rate (%) ↑ | 100.00 (0.00) | 28.69 (3.88) | 44.19 (5.23) | 2.22 (1.19) | 1.97 (0.88) | 2.11 (0.84) | 46.68 (4.51) | 2.19 (1.21) | 2.19 (1.21) | 2.19 (1.21) | 14.04 (2.52) | 13.49 (2.57) | 13.83 (2.40) | 2.19 (1.21) | 2.19 (1.21) | 2.19 (1.21) | 3.78 (1.52) | 2.11 (0.80) | 2.19 (1.21) | 2.19 (1.21) | 2.19 (1.21) | 2.19 (1.21) | 99.50 (6.29) | 4.45 (2.30) | 3.89 (2.23) | 2.19 (1.21) | 2.19 (1.21) | 2.19 (1.21) | 99.70 (4.71) | 34.10 (4.33) | 2.48 (0.81) |
| **XLarge** | Reward ↑ | 54.27 (5.19) | 57.44 (5.21) | 80.32 (6.61) | 2.80 (1.17) | 1.70 (1.00) | 1.80 (0.87) | 102.35 (5.16) | 2.80 (1.17) | 2.80 (1.17) | 2.80 (1.17) | 26.60 (2.33) | 25.50 (2.50) | 26.20 (2.44) | 2.80 (1.17) | 2.80 (1.17) | 2.80 (1.17) | 3.94 (1.77) | 1.80 (1.25) | 2.80 (1.17) | 2.80 (1.17) | 2.80 (1.17) | 2.80 (1.17) | 196.43 (1.56) | 4.20 (2.18) | 4.50 (1.75) | 2.80 (1.17) | 2.80 (1.17) | 2.80 (1.17) | 195.73 (1.61) | 64.61 (5.46) | 2.80 (0.87) |
|  | Feasible (%) ↑ | 27.05 (2.62) | 28.60 (2.62) | 39.80 (3.39) | 0.90 (0.58) | 0.35 (0.50) | 0.40 (0.44) | 50.80 (2.54) | 0.90 (0.58) | 0.90 (0.58) | 0.90 (0.58) | 12.80 (1.17) | 12.25 (1.25) | 12.60 (1.22) | 0.90 (0.58) | 0.90 (0.58) | 0.90 (0.58) | 1.70 (0.90) | 0.40 (0.62) | 0.90 (0.58) | 0.90 (0.58) | 0.90 (0.58) | 0.90 (0.58) | 98.20 (0.78) | 1.60 (1.09) | 1.75 (0.87) | 0.90 (0.58) | 0.90 (0.58) | 0.90 (0.58) | 97.85 (0.81) | 32.25 (2.75) | 0.90 (0.44) |
|  | Optimality Rate (%) ↑ | 100.00 (0.00) | 106.87 (14.64) | 148.77 (13.11) | 5.33 (2.62) | 3.24 (2.08) | 3.39 (1.76) | 190.06 (18.29) | 5.33 (2.62) | 5.33 (2.62) | 5.33 (2.62) | 49.35 (5.67) | 47.36 (6.28) | 48.63 (6.03) | 5.33 (2.62) | 5.33 (2.62) | 5.33 (2.62) | 7.26 (3.06) | 3.44 (2.58) | 5.33 (2.62) | 5.33 (2.62) | 5.33 (2.62) | 5.33 (2.62) | 365.09 (33.34) | 7.83 (3.94) | 8.35 (3.17) | 5.33 (2.62) | 5.33 (2.62) | 5.33 (2.62) | 363.73 (32.54) | 119.87 (12.94) | 5.29 (2.05) |

Table 4: Performance across different seeds. GNN-based models trained with reinforcement learning are sensitive to initial weights. TARGETNET, however, consistently outperforms the ablation models on median reward and optimality, particularly at larger scales. It also significantly reduces variance compared to the brittle TARGETNET, demonstrating greater robustness to initialization and the ability to learn policies that generalize better.

## Appendix H Demonstration

We deploy our TARGETNET algorithm in the Robotarium environment [97, 98] to validate its effectiveness in real-world robotic task execution scenarios. In this setup, a fleet of robots autonomously navigates toward assigned task locations, efficiently executing each task before transitioning to subsequent assignments. The Robotarium platform enables safe, scalable, and repeatable experiments, allowing us to visually demonstrate TARGETNET's ability to coordinate multi-robot systems in completing a sequence of spatially distributed tasks with minimal conflict and optimal coverage.

While TARGETNET efficiently assigns and sequences tasks, it is important to note that the underlying scheduler does not explicitly account for multi-agent collisions during path planning. As a result, robots may experience temporary delays when navigating through congested areas or when encountering other agents along their paths. These interactions can lead to deviations from the expected travel times computed by the scheduler, particularly in high-density task scenarios. Despite this, TARGETNET maintains robust task completion and overall system efficiency, as dynamic collision handling by the Robotarium platform mitigates potential conflicts in real time. This highlights a potential avenue for future improvements by integrating collision-aware path planning directly into the task scheduler as presented in Section 6

## Appendix I Code Access

Our code can be found in `https://github.com/CORE-Robotics-Lab/NeurIPS2025_TARGETNET`

