# OpenReview forum: "Heterogeneous Graph Transformers for Simultaneous Mobile Multi-Robot Task Allocation and Scheduling under Temporal Constraints"
_NeurIPS.cc/2025/Conference — NeurIPS 2025 poster_

### Official Review · Reviewer_nWbH · 2025-06-18

**Clarity:** 3
**Significance:** 1
**Originality:** 2
**Rating:** 3
**Confidence:** 3

**Summary:**

This paper aims to improve the scalability of the solver in multi-agent task allocation and scheduling domain under constraints. The authors follow previous GNN-based methods and propose TARGETNET, which treats agents, tasks, and allocation all as nodes in a graph and relations between them as edges. In each time step, the allocation result is decided via message passing through the graph. The authors further conducted a mass of experiments against multiple baselines to show the effect of TARGETNET.

**Questions:**

1. Besides the agents and tasks as nodes, the authors further include $|A||T|$ assignment nodes in the graph. When encountered with extremely large scale tasks, the complexity of the graph would increase significantly. However, the experiment results show that the computation time of TARGETNET is lower than baselines. Could the authors explain why TARGETNET maintains a high efficiency?

2. As shown in Figure 3 and 4, the performance of TARGETNET is severely declined after removing the residual connections. Is this indicating that the improved performance of TARGETNET is due to the residual connections rather than the graph-based learning methodology?

**Ethical Concerns:**

["NO or VERY MINOR ethics concerns only"]

**Final Justification:**

I have updated my assessment to ``3: borderline reject''. The authors have described a well-designed framework for multi-agent task allocation, but the experiment results indicate that the key to the improved performance is the residual connection in the network rather than the edge attention proposed in Section 3.4. This gives the impression that the proposed method looks fancy in Figure 2 but lacks practical utility. During the rebuttal period, the author modified TARGETNET and proposed an improved version TARGETNETv2, which is much stronger but needs to be properly integrated into the paper with further ablation studies.

**Limitations:**

Yes but not sufficient. Please see weaknesses and questions.

**Paper Formatting Concerns:**

No formatting concerns.

**Quality:**

2

**Strengths And Weaknesses:**

**Strengths**:
1. The presentation of the paper is very detailed. The authors have provided various materials to illustrate the methodology and the results, including graphs, equations, pseudo-code, and videos.
2. The authors conducted extensive experiments, employing a variety of baselines and designing multiple metrics to evaluate the performances. All these lead to convincing results.

**Weaknesses**:
1. There are some expression and grammar mistakes in the paper (e.g., line 51, 163 and 181), which should be avoided in academic writing.
2. Overall, the performance of TARGETNET is not outstanding enough compared to the baselines. It exhibits clear superiority in the computation time, but its optimal and successful rates are inferior to MILP solver in small, medium and large scale tasks. As MILP solver is supported by solid mathematical theory, users would prefer MILP in most cases, making the application of TARGETNET limited in extremely large cases.

---

> ### Author Rebuttal · Authors · 2025-07-30
>
> We thank the reviewer for their detailed feedback and constructive comments. We have made revisions based on the suggestions.
>
> > **There are some expression and grammar mistakes in the paper (e.g., line 51, 163 and 181), which should be avoided in academic writing.**
>
> We have corrected the grammar mistakes mentioned and further improved our manuscript.
>
> > **Overall, the performance of TARGETNET is not outstanding enough compared to the baselines. It exhibits clear superiority in the computation time, but its optimal and successful rates are inferior to MILP solver in small, medium and large scale tasks. As MILP solver is supported by solid mathematical theory, users would prefer MILP in most cases, making the application of TARGETNET limited in extremely large cases.**
>
> We would like to thank the reviewer for their thorough analysis of the limitations of the Machine Learning-based approaches to optimization problems.
>
> Our primary contribution is not to replace MILP in scenarios where it is feasible, but to provide a scalable solution that delivers high-quality, near-optimal schedules in a fraction of the time for problem sizes where MILP fails entirely. For instance, our experiments show that TARGETNET finds solutions orders of magnitude faster, making it practical for real-time applications where a good solution quickly is more valuable than a perfect solution that arrives too late.
>
> Developing fast, high-quality approximate solvers like TARGETNET is a vital research direction that can work together with exact solvers. As noted in [1, 2], such methods can provide a high-quality "warm start" to dramatically reduce the search space and runtime of a MILP solver. Therefore, we believe TARGETNET is not "limited" but rather is a significant and practical step forward for a broad and critical class of computationally challenging optimization problems.
>
> We have also conducted additional experiments, including TARGETNET without edge attention, TARGETNET\E, and a modification of TARGETNET with a non-normalized edge attention. The results, presented as part of the rebuttal for Reviewer EqLo's comments, show that one of the three seeds is able to converge into a policy that is within 5% of the optimal fully feasible solution on an extra-large scale. The modification of TARGETNETv2 allows a more robust model that can, on average, converge to a policy that is better than other learning based baselines, indicating a less brittle model that depends on the initial model weights. These results will be added to the camera-ready version.
>
> [1] Bengio, Y., Lodi, A., & Prouvost, A. (2021). Machine learning for combinatorial optimization: a methodological tour d’horizon. _European Journal of Operational Research_, _290_(2), 405-421.
> [2] Kruber, M., Lübbecke, M. E., & Parmentier, A. (2017, May). Learning when to use a decomposition. In _International conference on AI and OR techniques in constraint programming for combinatorial optimization problems_ (pp. 202-210). Cham: Springer International Publishing.
>
> > **Besides the agents and tasks as nodes, the authors further include |A||T| assignment nodes in the graph. When encountered with extremely large scale tasks, the complexity of the graph would increase significantly. However, the experiment results show that the computation time of TARGETNET is lower than baselines. Could the authors explain why TARGETNET maintains a high efficiency?**
>
> TARGETNET only constructs a single graph representation of the world, including the heterogenous travel time from one task to another task for each agent, which is represented as an edge between the Assignment Nodes, with a complexity of $O(|A||T|^2)$ number of edges due to the limitation of per agent travel.
>
> Unlike baselines that still require the simulation of the agent movement to new location to update agent makespan and when they would be available to move to the next task location, the simultaneous nature of TARGETNET allow for the generation of the entire schedule without needing to interact with the environment, relying on the initial state observation, $s_0$, to generate a full schedule.
>
> Furthermore, TARGETNET scales polynomially with the number of agents and tasks while the problem itself is NP-hard. This is one of the key advantages of learning-based models, as they can learn potential heuristics that outperform existing methods.
>
> > **As shown in Figure 3 and 4, the performance of TARGETNET is severely declined after removing the residual connections. Is this indicating that the improved performance of TARGETNET is due to the residual connections rather than the graph-based learning methodology?**
>
> We would like to refer the Reviewer to the rebuttal to the feedback of Reviewer EqLo for additional experiments for our model without edge attention, TARGETNET\E. We agree with the reviewer that the performance of the Graph Models has a significant increase in the performance with the addition of Residual Connections, and the presence of the Edge Features makes the model less brittle, capable of learning a better mean performance across 3 different seeds (10, 11, 12).
>
> Furthermore, our additional experiments with TARGETNETv2 that allow a non-normalized edge attention encapsulate the policies that can be learned by TARGETNET\E, as they are able to learn the same policies. This makes the addition of Edge Attention lead to a more robust model that can learn a wide range of policies that models without edge attention are not able to represent.
>
> We are currently running the training and evaluation of the Sequential variant of the model without edge features, Seq-TARGETNET\E. Due to the slower nature of the model, we will provide these results in the camera-ready version of our work along with a detailed discussion.

---

> > ### Comment · Reviewer_nWbH · 2025-08-03
> >
> > The reviewer appreciates the sophisticated replies of the authors. The further introduction of the motivation of TARGETNET has addressed some of my concerns, but I still have questions about the residual connections:
> > 1. The performance of TARGETNET/E degrades when running on multiple random seeds, is this because removing edge attention has reduced the model’s expressive capacity?
> > 2. Why TARGETNET/E still outperforms TARGETNET in large and extremely large tasks?
> > 3. As the application of TAGGETNET is mainly at tasks with large scales, does it mean edge attention is useless, since TARGETNET without edge attention has a simpler implementation and better performance on large tasks?

---

> > > ### Author Response · Authors · 2025-08-05
> > >
> > > We thank the reviewer for their follow-up questions. For clarity in this discussion, we will refer to the three models as follows:
> > >
> > > **TARGETNET\E:** does not account for edge features.
> > >
> > > **TARGETNET:** Uses a softmax function across multi-head attention that always accounts for edge features.
> > >
> > > **TARGETNETv2:** Removes the softmax from TARGETNET, allowing the model to express capabilities of both TARGETNET\E and TARGETNET. On average, TARGETNETv2 has a better performance than TARGETNET\E and TARGETNET, as it is less brittle and more expressive.
> > >
> > > > **Response to Q1:**
> > >
> > > Yes, the reviewer is correct. The performance degradation of TARGETNET\E across different seeds is a direct consequence of its brittleness, which stems from its inability to leverage edge features.
> > >
> > > While one specific seed of TARGETNET\E might get "lucky" with its initialization and find an effective policy that ignores edge data, it is not a reliable strategy. This is precisely the problem that TARGETNETv2 is designed to solve.
> > >
> > > Our focus is on creating a model that performs consistently well without depending on luck. TARGETNETv2 achieves this by performing at or near the level of TARGETNET\E's best-case scenario. Furthermore, TARGETNETv2 consistently outperforms TARGETNET\E on average across multiple seeds as shown in Table 1 below, while drastically reducing the performance variance across different initializations as shown in Table 2 below. This demonstrates that including the edge feature into the model offers a greater robustness and scalability that is a direct result of its ability to leverage edge features.
> > >
> > > | **Scale**|**Data Type**|**MILP Solver (reference)**|**TARGETNET\E**|**TARGETNET (Ours)**|**TARGETNET_v2 (Ours)** |
> > > |-|-|-|-|-|-|
> > > | **Small** | **Mean Reward** |20.08±0.05|8.73±7.79|11.46±4.06|**11.76±6.28** |
> > > | **Medium** | **Mean Reward** |50.06±0.03|19.07±21.16|**24.62±8.45**|22.30±19.50 |
> > > | **Large** | **Mean Reward** |98.31±4.54|35.24±44.08|34.49±22.61|**44.57±39.72** |
> > > | **XLarge** | **Mean Reward** |54.27±5.19|68.38±90.55|50.21±43.91|**87.71±80.44** |
> > >
> > > _TABLE 1:  Mean performance comparison between TARGETNET\E, TARGETNET and TARGETNETv2 across three seeds showing that the addition of edge attention improves the average performance as a result of more robust models that account for edge features._
> > >
> > > | **Scale**|**Small**|**Medium**|**Large**|**XLarge**|
> > > |-|-|-|-|-|
> > > | **Variance Reduction of Mean Reward (%)**|35.01032|15.07454|18.80387|21.08361|
> > >
> > > _TABLE 2: Variance Reduction of Mean Reward TARGETNETv2 compared to TARGETNET\E._
> > >
> > > > **Response to Q2:**
> > >
> > > The original TARGETNET uses a softmax function, which forces the model to consider edge features in every decision. As the performance of the models in large and extra-large scales show, the optimal policy might be simpler and may not require strict attention to edge features.
> > >
> > > This reveals two key insights:
> > >
> > > - A model that cannot ignore edge features (TARGETNET) is too restrictive.
> > >
> > > - A model that cannot use edge features (TARGETNET\E) is too brittle.
> > >
> > > TARGETNETv2 solves this dilemma. By removing the softmax across attention heads, TARGETNETv2 gains the flexibility to learn when to use edge features and when to ignore them. It captures the best of both approaches: it can learn the simple, high-performing policy of a "lucky" TARGETNET\E seed but can also leverage edge features to find even better, more stable policies that TARGETNET\E is incapable of discovering.
> > >
> > > The result is a single, robust model that outperforms both predecessors not by inflating best-case numbers, but by consistently achieving strong results.
> > >
> > > > **Response to Q3:**
> > >
> > > Over an average of 3 seeds, TARGETNET\E and TARGETNET both perform worse than TARGETNETv2, which indicates that edge attention is not useless, as TARGETNETv2 is able to learn different policies that perform well across the board.
> > >
> > > The performance of the different policies indicate that the presence of edge attention can lead to better policies on average that scale better that TARGETNET without edge attention.
> > >
> > > While removing edge attention makes the model simpler, a trade-off is made between the increase in brittleness of the model and minor linear performance cost, even in large scale problems.
> > >
> > > Edge attention can therefore lead to more robust models that can learn better policies on average.
> > >
> > > - TARGETNETv2 consistently outperforms both models on median reward and optimality at larger scales.
> > >
> > > - TARGETNETv2 significantly reduces the variance compared to the brittle TARGETNET\E, making it far more reliable.
> > >
> > > In conclusion, edge attention is not only useful—it is essential for building a model that is both high-performing and dependable. TARGETNETv2 provides the architectural framework to do so effectively. We have revised the manuscript to incorporate this narrative and the clarified results.

---

> > > > ### Comment · Reviewer_nWbH · 2025-08-08
> > > >
> > > > Thanks the authors for the detailed reply. The conclusion that the version without softmax performs better than that with softmax is quite contrary to common intuition, so I remain somewhat skeptical about this conclusion. First, softmax can provide good numerical stability for the network, so will it cause instability and Inf or NaN values during training if we remove softmax? Second, by referring to the structure of Transformer, we can find it is also implemented with softmax function. Why does the softmax work well in Transformer but restrict the performance of TARGETNET?
> > > >
> > > > As there is not much time remaining for discussion, I would like to ask the authors to offer a theoretical explanation instead of conducting further experiments. Thanks again for the authors patience and efforts.

---

> > > > > ### Author Response · Authors · 2025-08-08
> > > > >
> > > > > > **The conclusion that the version without softmax performs better than that with softmax is quite contrary to common intuition, so I remain somewhat skeptical about this conclusion.**
> > > > >
> > > > > Softmax forces nonzero attention on all inputs; in our setting, this wastes capacity and hurts performance.
> > > > >
> > > > > The softmax function, defined as $\text{softmax}(z_i)​=\frac{e^{z_i}}{\sum_{j}{e^{z_j}}}​​$, normalizes scores into a probability distribution where every output is strictly positive. This is suboptimal for our setting because it limits the model's expressive capabilities by preventing zero-attention. By allowing zero-attention, TARGETNET can learn to completely ignore irrelevant edge features, a critical ability for graph neural networks where multiple edges converge on a single target node.
> > > > >
> > > > > The ability of having zero-attention is supported by previous research [1], demonstrates that enabling a model to disregard certain inputs improves its expressive capacity and performance.
> > > > >
> > > > > We will expand our related work section to address this limitation of softmax.
> > > > >
> > > > > > **First, softmax can provide good numerical stability for the network, so will it cause instability and Inf or NaN values during training if we remove softmax?**
> > > > >
> > > > > Removing softmax does not cause instability in our model. We employ a combination of established methods to ensure numerical stability and prevent `Inf` or `NaN` values during training.
> > > > >
> > > > > - **Input Normalization:** We normalize all inputs, which prevents node and edge embeddings from growing exponentially across the network's layers. This scaling is crucial for maintaining stable embedding values.
> > > > >
> > > > > - **Gradient Clipping:** We clip gradients by setting their L2-norm to 1. This directly stops the model's weights from growing uncontrollably, which is a primary cause of exploding gradients and resulting `Inf`/`NaN` values.
> > > > >
> > > > > - **L2 Regularization:** This technique penalizes large weights, discouraging the model from converging to sharp minima and further preventing exploding gradients.
> > > > >
> > > > > The combination of these methods is highly effective. Our empirical results confirm that TARGETNETv2 does not suffer from instability, and we observed zero occurrences of `Inf` or `NaN` values during training.
> > > > >
> > > > > > **Second, by referring to the structure of Transformer, we can find it is also implemented with softmax function. Why does the softmax work well in Transformer but restrict the performance of TARGETNET?**
> > > > >
> > > > > The role of attention differs fundamentally between language modeling and combinatorial optimization. Language models require softmax to capture broad, distributed contextual dependencies across a sequence. In contrast, our combinatorial optimization task benefits from focused attention where the model can exploit explicit node and edge features for decision-making. The strict positivity of softmax is an unnecessary constraint that reduces the model's ability to disregard irrelevant elements.
> > > > >
> > > > > While softmax is standard in Transformers, it is not universally essential for high performance. Ongoing research demonstrates that alternatives like linear attention [2] and even replacing attention layers entirely with feedforward networks [3] can be effective. In fact, our approach of removing softmax aligns with the high-performing "Attention Separate Heads Layer Replacement" (ASLR) configuration in [3], which outperforms other methods on benchmark tasks.
> > > > >
> > > > > We will expand our related work to include this comparison.
> > > > >
> > > > > [1] E. Miller, “Attention Is Off By One,” _Evanmiller.org_, 2023. https://www.evanmiller.org/attention-is-off-by-one.html (accessed Aug. 08, 2025).
> > > > >
> > > > > [2] Han, D., Pu, Y., Xia, Z., Han, Y., Pan, X., Li, X., ... & Huang, G. (2024). Bridging the divide: Reconsidering softmax and linear attention. _Advances in Neural Information Processing Systems_, _37_, 79221-79245.
> > > > >
> > > > > [3] Bozic, V., Dordevic, D., Coppola, D., Thommes, J., & Singh, S. P. (2023). Rethinking attention: Exploring shallow feed-forward neural networks as an alternative to attention layers in transformers. _arXiv preprint arXiv:2311.10642_.

---

> > > > > > ### Comment · Reviewer_nWbH · 2025-08-09
> > > > > >
> > > > > > Thanks for the authors' reply. The version of TARGETNET in the paper is not significant in performance and applicability, and I suggest the authors implement substantial revisions to the manuscript by replacing TARGETNET with v2 version. I will also update my assessment accordingly.

---

> > > > > > > ### Author Response · Authors · 2025-08-09
> > > > > > >
> > > > > > > We thank the reviewer for their questions and feedback. We will integrate the TARGETNETv2 into the final version of the manuscript.

---

### Official Review · Reviewer_6ugj · 2025-07-01

**Clarity:** 3
**Significance:** 3
**Originality:** 3
**Rating:** 5
**Confidence:** 3

**Summary:**

This paper proposes TARGETNET, a reinforcement learning (RL)-based framework for heterogeneous mobile multi-agent task allocation and scheduling problems (HM-MATAS). TARGETNET tackles HM-MATAS by formulating it as an RL problem with graph input, and then uses REINFORCE to optimize the policy. To better handle the graph input, the authors propose a new graph transformer with node and edge attention mechanisms. Along with residual connections, this graph transformer has zero-shot generalization to problems with larger scale. The proposed framework is evaluated on tasks with up to 40 agents and 200 tasks, and the results show that TARGETNET outperforms existing baselines.

**Questions:**

1. Please provide more discussion on the difference in problems and methods between this paper and [4,5,6].
2. It is observed that the sequential decision making version of TARGETNET have better performance than the one-step version. Some prior work [7] has shown the potential to boost the performance of stateless policy by a simple distillation process, which means we can distill the sequential policy to a one-step policy by a simple behavior cloning process. Can you try this simple distillation process to boost the performance of the one-step version of TARGETNET?
3. Can you discuss the time cost of training stage of TARGETNET?
4. The edge attention mechanism to incorporate the edge information seems to be an important contribution of this paper. However, there is a simple way to incorporate the edge information into the node attention mechanism: we can create a new node for each edge with the edge information as the node feature. This way, we can use the same node attention mechanism to incorporate the edge information. Can you discuss the difference between this simple way and the proposed edge attention mechanism? And what if the baseline method Res-HetGAT uses this simple way to incorporate the edge information?

[7] Ye, Jianing, Chenghao Li, Jianhao Wang and Chongjie Zhang. “Towards Global Optimality in Cooperative MARL with the Transformation And Distillation Framework.” (2022).

**Ethical Concerns:**

["NO or VERY MINOR ethics concerns only"]

**Final Justification:**

As most of my concerns have been clarified by the authors' response, I will keep my current rating.

**Limitations:**

Yes

**Quality:**

3

**Strengths And Weaknesses:**

**Strengths**
1. The presentation of this paper is clear and well-organized.
2. HM-MATAS is an important problem in the field of multi-agent systems and complexity theory.
3. The proposed framework is novel and has a good performance on the benchmark tasks, especially on the larger scale tasks.
4. This framework demonstrates how RL and GNN can be combined to effectively address large-scale NP-hard problems involving both task allocation and scheduling over graph-structured representations.


**Weaknesses**
1. The proposed method has a fast inference speed, while the training time is not mentioned.
2. The interpretability of the proposed method is limited, which is a common problem in deep RL.
3. Using RL to solve combinatorial optimization problems (e.g. graph-based problems) is not a new idea [1,2,3]. The authors should discuss more related work in this area [4,5,6].

[1] Wagner, Adam Zsolt. “Constructions in combinatorics via neural networks.” ArXiv abs/2104.14516, 2021.

[2] Darvariu V A, Hailes S, Musolesi M. Graph reinforcement learning for combinatorial optimization: A survey and unifying perspective[J]. arXiv preprint arXiv:2404.06492, 2024.

[3] Peng Y, Choi B, Xu J. Graph learning for combinatorial optimization: a survey of state-of-the-art[J]. Data Science and Engineering, 2021.

[4] Ma, Qiang, Suwen Ge, Danyang He, Darshan D. Thaker and Iddo Drori. “Combinatorial Optimization by Graph Pointer Networks and Hierarchical Reinforcement Learning.” ArXiv abs/1911.04936, 2019.

[5] Batuhan Altundas, Zheyuan Wang, Joshua Bishop, and Matthew Gombolay. Learning Coor- dination Policies over Heterogeneous Graphs for Human-Robot Teams via Recurrent Neural Schedule Propagation.

[6] Zheyuan Wang and Matthew Gombolay. Stochastic Resource Optimization over Heterogeneous Graph Neural Networks for Failure-Predictive Maintenance Scheduling.

---

> ### Author Rebuttal · Authors · 2025-07-30
>
> We sincerely thank the reviewer for their detailed feedback and insightful comments. We have incorporated revisions based on this feedback.
>
> > **The proposed method has a fast inference speed, while the training time is not mentioned.**
> > **Can you discuss the time cost of training stage of TARGETNET?**
> The training time was not mentioned as each model is trained once in the small scale and tested on a range of different scales.
>
> We have conducted an additional evaluation on our system for learning-based baselines for over 100 epochs. We report the mean and standard deviation of time in seconds to train each model for a single epoch below:
>
> |  | **Mean Training Time (s)** | **Stdev Training Time** |
> |---|---|---|
> | **HetGAT** | 28.310 | 1.030 |
> | **Res-HetGAT** | 29.735 | 0.972 |
> | **Seq-TARGETNET/ER** | 43.628 | 1.838 |
> | **Seq-TARGETNET/R** | 66.129 | 3.395 |
> | **Seq-TARGETNET** | 61.843 | 2.028 |
> | **TARGETNET/ER** | 2.166 | 0.085 |
> | **TARGETNET/R** | 2.998 | 0.095 |
> | **TARGETNET/E** | 2.206 | 0.080 |
> | **TARGETNET** | 3.039 | 0.103 |
>
> As seen in the training time, the training speed of different models is consistent with the inference speed. Simultaneous decision-making models of TARGETNET are faster by a factor of x20. As the models are trained on 20 task problems, the empirical results align with the theoretical complexity of $O(|A||T|^2)$ for sequential models and $O(|A||T|) for simultaneous models. The increase in the feature space size with the presence of residuals and edge attention leads to an increase in training and evaluation time.
>
> The training time presented above also implies that it is faster to train a simultaneous TARGETNET over the existing sequential models from prior work, leading to training and deploying models within the time frame that it takes an exact solver to produce a single suboptimal solution for the extra-large scale.
>
> > **The interpretability of the proposed method is limited, which is a common problem in deep RL.**
>
> We agree with the Reviewer that Interpretability is a common problem in both Deep RL and Optimization Problems. We further discuss it in paragraph starting in Line 329 in Section 6.
>
> > **a. Using RL to solve combinatorial optimization problems (e.g. graph-based problems) is not a new idea [1,2,3].**
>
> We agree that there are prior works that focus on using RL to solve combinatorial optimization problems. The method presented in TARGETNET, which uses RL for both Task Allocation and Scheduling in a single forward propagation, is novel.
>
> > **Please provide more discussion on the difference in problems and methods between this paper and [4,5,6].**
>
> We would like to thank the reviewer for referring the related work which we will discuss more in depth. We agree that contextualizing our work within the broader field of RL for Combinatorial Optimization (CO) is crucial. We have revised our related work section to better position TARGETNET and now include a detailed discussion of the suggested papers. The key distinctions are:
>
> - **[4] Ma et al., 2019:** This work proposes a hierarchical GNN for the Traveling Salesman Problem with Time Windows (TSPTW). While related, our HM-MATAS formulation addresses a more complex problem, equivalent to a multi-agent TSP variant (m-TSPTW) with additional heterogeneity constraints for agents and tasks. TARGETNET is designed to handle the joint allocation and scheduling aspects of this multi-agent, multi-task setting in a single pass, and accounts for heterogeneous task durations along with the heterogeneous travel time.
>
> - **[5] Altundas et al. & [6] Wang and Gombolay:** The problem formulation of both [5] and [6] assumes that travel times are negligible compared to task service durations, leading to a simpler graph representation of the problem. Our framework, TARGETNET, is explicitly designed to handle scenarios where heterogeneous travel times are a critical component of the scheduling decision and may be a key factor in success. Furthermore, [5] relies on an ensemble of policies to achieve its best performance, whereas TARGETNET generates a high-quality schedule in a single forward pass. [6] focuses on a stochastic maintenance scheduling problem, leveraging the HetGAT model we benchmark against. However, the key challenge in [6] is _when_ to dispatch agents to tasks that appear over time, which differs from our deterministic generation of complete schedules in fully observable settings.
>
> We have incorporated these comparisons into our revised manuscript.
>
> > **It is observed that the sequential decision making version of TARGETNET have better performance than the one-step version. Some prior work [7] has shown the potential to boost the performance of stateless policy by a simple distillation process, which means we can distill the sequential policy to a one-step policy by a simple behavior cloning process. Can you try this simple distillation process to boost the performance of the one-step version of TARGETNET?**
>
> The proposed distillation is indeed a powerful technique for creating a fast and high-performance policy. However, the primary motivation for our one-step model is to demonstrate its capability to learn through Reinforcement Learning. The goal is to provide a lightweight alternative for scenarios with limited computational budgets. The distillation process, while effective, would first require training the full sequential model, thereby negating the training-time advantages that make the one-step model appealing.
>
> Our paper's focus is on analyzing the trade-off between the computationally expensive sequential approach and the efficient one-step approach. While distilling the sequential policy is an excellent direction for future work to create a fast inference model, it addresses a different research question than the one we focus on.
>
> Furthermore, the additional experiments conducted, as presented in the rebuttal for Reviewer EqLo's comments, show that simultaneous models are able to learn policies that outperform sequential models in every scale while being more robust to initial starting weights.
>
> > **The edge attention mechanism to incorporate the edge information seems to be an important contribution of this paper. However, there is a simple way to incorporate the edge information into the node attention mechanism: we can create a new node for each edge with the edge information as the node feature. This way, we can use the same node attention mechanism to incorporate the edge information. Can you discuss the difference between this simple way and the proposed edge attention mechanism? And what if the baseline method Res-HetGAT uses this simple way to incorporate the edge information?**
>
> We thank the reviewer for their suggested feedback. While the edge features can be represented as a node feature with the addition of new nodes for each node, the added node would lead to the need for additional layers for message passing. Furthermore, the two-hop message passing for the node features would lead to added complexities and potential for information loss. Our proposed method also combines the edge and node features when updating the node features of the next layer, leveraging the linearity of attention such that $A(B || C) = A_1 B + A_2 C$, where $A = (A_1||A_2)$, for attention vectors, $A$, $A_1$ and $A_2$ and feature vectors $B$ and $C$.
>
> For both HetGAT and HGT, the simple way of incorporating the edge features and attention would lead to increased complexity and need for additional layers, leading to increased computation time that may surpass the method proposed in our work.
>
> Based on the feedback of the reviewer, we have also improved our model, adding TARGETNETv2 into our results. TARGETNETv2 removes the softmax of attention heads in the edge feature, allowing for our model to represent models with and without edge attention by learning the attention value. This leads to a more versatile and robust model in the presence of relational information, such as heterogeneous travel times. Again, we would like to refer the reviewer to the rebuttal for Reviewer EqLo's comments for these additional results.
>
> [1] Wagner, Adam Zsolt. “Constructions in combinatorics via neural networks.” ArXiv abs/2104.14516, 2021.
>
> [2] Darvariu V A, Hailes S, Musolesi M. “Graph reinforcement learning for combinatorial optimization: A survey and unifying perspective” [J]. arXiv preprint arXiv:2404.06492, 2024.
>
> [3] Peng Y, Choi B, Xu J. “Graph learning for combinatorial optimization: a survey of state-of-the-art”[J]. Data Science and Engineering, 2021.
>
> [4] Ma, Qiang, Suwen Ge, Danyang He, Darshan D. Thaker and Iddo Drori. “Combinatorial Optimization by Graph Pointer Networks and Hierarchical Reinforcement Learning.” ArXiv abs/1911.04936, 2019.
>
> [5] Batuhan Altundas, Zheyuan Wang, Joshua Bishop, and Matthew Gombolay. “Learning Coordination Policies over Heterogeneous Graphs for Human-Robot Teams via Recurrent Neural Schedule Propagation.”
>
> [6] Zheyuan Wang and Matthew Gombolay. “Stochastic Resource Optimization over Heterogeneous Graph Neural Networks for Failure-Predictive Maintenance Scheduling”.

---

> > ### Comment · Reviewer_6ugj · 2025-08-07
> >
> > Thank you for the response.  I have carefully reviewed the authors' rebuttal and will consider it during the final decision process.

---

### Official Review · Reviewer_EqLo · 2025-07-03

**Clarity:** 3
**Significance:** 2
**Originality:** 2
**Rating:** 4
**Confidence:** 2

**Summary:**

The paper proposes TARGETNET, a scalable one-shot framework for simultaneous task allocation and scheduling in heterogeneous multi-agent systems under temporal constraints. By modeling the problem as a heterogeneous graph and using a Transformer with edge-aware attention, it efficiently captures agent-task relationships, travel times, and scheduling dependencies. Trained via reinforcement learning, TARGETNET generalizes from small to large-scale problems, outperforming heuristics, metaheuristics, and prior GNNs in both solution quality and inference speed.

**Questions:**

1. There are ablation studies on TARGETNET\ER and TARGETNET\R, how about TARGETNET\E?

2. Why does sequential version performs better on the largest task? What are the typical failure modes of TARGETNET under this setting?

**Ethical Concerns:**

["NO or VERY MINOR ethics concerns only"]

**Limitations:**

yes

**Quality:**

3

**Strengths And Weaknesses:**

Strengths
1. Simultaneous task allocation and scheduling in a single forward pass — faster and more scalable than sequential or optimization-based methods.
2. Edge-aware Heterogeneous Graph Transformer effectively captures complex agent-task relationships, including travel time, execution time, and temporal constraints.
3. Strong generalization from small-scale training to large-scale deployment without retraining, with up to 250× speedup over baselines and near-optimal quality.

Weaknesses
1. Lack of interpretability — like most deep GNNs, the learned policy is hard to analyze or debug in safety-critical applications.
2. Slightly weaker performance than sequential models in extremely large-scale cases, which can benefit from per-step re-optimization (but at a huge cost in time).
3. The design of transformer seems very specific to the nature of MATAS. I wish to see a more principled design like incorperating next-token prediction, and test the scaling law, etc.

---

> ### Author Rebuttal · Authors · 2025-07-30
>
> We thank the reviewer for their detailed feedback and constructive suggestions. Below, we address the noted weaknesses and questions.
>
> >  **Lack of interpretability — like most deep GNNs, the learned policy is hard to analyze or debug in safety-critical applications.**
>
> We acknowledge that interpretability is a limitation of our model, and discuss it in the limitations section. Future Work discusses potential approaches to improve the interpretability while leveraging the advantages of the model.
>
> > **Slightly weaker performance than sequential models in extremely large-scale cases, which can benefit from per-step re-optimization (but at a huge cost in time).**
>
> In Section 6, we propose potential future work in utilizing a meta-policy that combines the performance of the sequential models with simultaneous models. Furthermore, we have conducted additional experiments, leading to increase in performance of our model TARGETNETv2 to near optimal levels while being less brittle than the models not using edge attention. These results can be seen below.
>
> > **The design of transformer seems very specific to the nature of MATAS. I wish to see a more principled design like incorporating next-token prediction, and test the scaling law, etc.**
>
> The suggestion of next-token prediction is rooted in autoregressive models like LLMs, which excel at sequential data. However, the Multi-Agent Task Allocation and Scheduling (MATAS) problem is fundamentally a combinatorial optimization problem on a graph, not a sequence generation task. While Transformers can be used in the context of combinatorial optimization problems [1], the approach of graph attention are applied to node and edge features that are connected to individual nodes on a graph, allow sparse convolution that can scale to larger problems [2].
>
> We would like to refer the reviewer to Hu et al. 2020 [3] for full explanation of the Heterogenous Graph Transformers (HGT) as well as Appendix 2. While the HGT leverage the attention mechanism popularized by Vaswani et al. 2017 [4], our model does not utilize token prediction in the same way that a transformer architecture would achieve.
>
> HGT is a common Graph Neural Network algorithm that leverages a distinct attention mechanism. To our knowledge, the edge attention mechanism we propose in our paper has not been integrated into the HGT architecture. While we leverage the method for the MATAS problems, further research would need to be conducted for any other problem where the relational edge features are encoded in the input of a model.
>
> We train our model in small scale problems and evaluate the trained models in large scale problem without any fine tuning, as such the presented results do not have a change in the parameter counts during training. Future work will involve training in different scale environments to test how the model training scales, however this is beyond the scope of this paper.
>
> > **There are ablation studies on TARGETNET\ER and TARGETNET\R, how about TARGETNET\E?**
>
> Below is the performance of the best of three seeds. Along with TARGETNET\E, we introduce TARGETNETv2, which is a modification of TARGETNET where edge attention is not normalized using the softmax function. The non-normalized attention allows for TARGETNETv2 to learn policies that can be learned by TARGETNET\E, by learning the edge attention weight to 0. The modification allows our model to learn:
>
> | **Scale** | **Data Type** | **MILP Solver** | **TARGETNET\E** | **TARGETNET (Ours)** | **TARGETNET (Ours) V2** |
> |-|-|-|-|-|-|
> | **Small** | **Max Reward** | 20.08±0.05 | **19.74±0.57** | 17.11±2.47 | _19.72±0.57_ |
> | **Small** | **Feasible (%)** | 100.00±0.00 | **98.28±2.85** | 84.88±12.47 | _98.17±2.88_ |
> | **Small** | **Optimality Rate (%)** | 100.00±0.00 | **98.31±2.82** | 85.18±12.30 | _98.21±2.86_ |
> | **Small** | **Optimality Gap** | 0.00±0.00 | **1.69±2.82** | 14.82±12.30 | _1.79±2.86_ |
> | **Medium** | **Reward** | 50.06±0.03 | **49.00±1.20** | 35.02±6.41 | _48.96±1.15_ |
> | **Medium** | **Feasible (%)** | 100.00±0.00 | **97.88±2.41** | 69.80±12.86 | _97.81±2.29_ |
> | **Medium** | **Optimality Rate (%)** | 100.00±0.00 | **97.88±2.40** | 69.96±12.81 | _97.81±2.29_ |
> | **Medium** | **Optimality Gap** | 0.00±0.00 | **2.12±2.40** | 30.04±12.81 | _2.19±2.29_ |
> | **Large** | **Reward** | 98.31±4.54 | _97.58±3.48_ | 59.16±10.86 | **97.81±1.76** |
> | **Large** | **Feasible (%)** | 98.27±4.55 | _97.53±3.48_ | 59.10±10.87 | **97.77±1.76** |
> | **Large** | **Optimality Rate (%)** | 100.00±0.00 | _99.50±6.29_ | 60.25±10.98 | **99.70±4.71** |
> | **Large** | **Optimality Gap** | 0.00±0.00 | _0.50±6.29_ | 39.75±10.98 | **0.30±4.71** |
> | **XLarge** | **Reward** | 54.27±5.19 | **196.43±1.56** | 109.04±10.03 | _195.73±1.61_ |
> | **XLarge** | **Feasible (%)** | 27.05±2.62 | **98.20±0.78** | 54.50±5.02 | _97.85±0.81_ |
> | **XLarge** | **Optimality Rate (%)** | 100.00±0.00 | **365.09±33.34** | 201.96±21.05 | _363.73±32.54_ |
> | **XLarge** | **Optimality Gap** | 0.00±0.00 | **-265.09±33.34** | -101.96±21.05 | _-263.73±32.54_ |
>
> Below is the mean and standard deviation of the performance of the learning-based models using three seeds (10, 11, 12) as an expansion of the ablation study presented in our paper.
>
> | **Scale** | **Data Type** | **MILP Solver (reference)** | **HetGAT** | **Res-HetGAT** | **Seq-TARGETNET\ER** | **Seq-TARGETNET\R** | **Seq-TARGETNET** | **TARGETNET\ER** | **TARGETNET\R** | **TARGETNET\E** | **TARGETNET (Ours)** | **TARGETNET_v2 (Ours)** |
> |-|-|-|-|-|-|-|-|-|-|-|-|-|
> | **Small** | **Mean Reward** | 20.08±0.05 | 12.13±3.25 | 5.39±3.75 | 2.10±0.00 | 2.10±0.00 | 7.81±6.65 | 2.11±0.01 | 2.52±0.26 | 8.73±7.79 | 11.46±4.06 | 11.76±6.28 |
> | **Small** | **Feasible (%)** | 100.00±0.00 | 58.66±18.14 | 22.93±20.06 | 5.53±0.00 | 5.53±0.00 | 37.08±34.65 | 5.57±0.06 | 7.67±1.29 | 40.23±41.05 | 55.28±21.93 | 57.25±32.57 |
> | **Small** | **Optimality Rate (%)** | 100.00±0.00 | 60.42±16.17 | 26.86±18.69 | 10.48±0.00 | 10.48±0.00 | 38.89±33.12 | 10.52±0.06 | 12.56±1.29 | 43.47±38.78 | 57.07±20.21 | 58.58±31.28 |
> | **Small** | **Optimality Gap** | 0.00±0.00 | 39.58±16.17 | 73.14±18.69 | 89.52±0.00 | 89.52±0.00 | 61.11±33.12 | 89.48±0.06 | 87.44±1.29 | 56.53±38.78 | 42.93±20.21 | 41.42±31.28 |
> | **Medium** | **Mean Reward** | 50.06±0.03 | 16.96±5.15 | 6.06±4.84 | 2.38±0.00 | 2.38±0.00 | 11.94±10.17 | 2.38±0.00 | 2.90±0.36 | 19.07±21.16 | 24.62±8.45 | 22.30±19.50 |
> | **Medium** | **Feasible (%)** | 100.00±0.00 | 32.24±10.61 | 10.12±9.67 | 2.76±0.00 | 2.76±0.00 | 22.75±20.38 | 2.76±0.00 | 3.82±0.72 | 36.81±43.19 | 48.47±17.71 | 43.59±39.72 |
> | **Medium** | **Optimality Rate (%)** | 100.00±0.00 | 33.88±10.29 | 12.11±9.66 | 4.75±0.00 | 4.75±0.00 | 23.86±20.31 | 4.75±0.00 | 5.79±0.71 | 38.10±42.27 | 49.18±16.89 | 44.55±38.95 |
> | **Medium** | **Optimality Gap** | 0.00±0.00 | 66.12±10.29 | 87.89±9.66 | 95.25±0.00 | 95.25±0.00 | 76.14±20.31 | 95.25±0.00 | 94.21±0.71 | 61.90±42.27 | 50.82±16.89 | 55.45±38.95 |
> | **Large** | **Mean Reward** | 98.31±4.54 | 34.24±11.94 | 11.90±12.76 | 2.13±0.00 | 2.13±0.00 | 22.00±20.36 | 2.13±0.00 | 2.78±0.21 | 35.24±44.08 | 34.49±22.61 | 44.57±39.72 |
> | **Large** | **Feasible (%)** | 98.27±4.55 | 33.53±12.27 | 10.91±12.77 | 1.13±0.00 | 1.13±0.00 | 21.48±20.44 | 1.13±0.00 | 1.80±0.19 | 34.57±44.52 | 34.11±23.02 | 44.13±40.08 |
> | **Large** | **Optimality Rate (%)** | 100.00±0.00 | 34.88±12.19 | 12.14±13.03 | 2.19±0.00 | 2.19±0.00 | 22.41±20.72 | 2.19±0.00 | 2.85±0.21 | 35.95±44.94 | 35.17±23.03 | 45.43±40.49 |
> | **Large** | **Optimality Gap** | 0.00±0.00 | 65.12±12.19 | 87.86±13.03 | 97.81±0.00 | 97.81±0.00 | 77.59±20.72 | 97.81±0.00 | 97.15±0.21 | 64.05±44.94 | 64.83±23.03 | 54.57±40.49 |
> | **XLarge** | **Mean Reward** | 54.27±5.19 | 70.31±25.58 | 27.07±33.44 | 2.80±0.00 | 2.80±0.00 | 63.38±85.53 | 2.80±0.00 | 2.63±0.12 | 68.38±90.55 | 50.21±43.91 | 87.71±80.44 |
> | **XLarge** | **Feasible (%)** | 27.05±2.62 | 34.82±12.98 | 13.07±16.76 | 0.90±0.00 | 0.90±0.00 | 31.35±43.00 | 0.90±0.00 | 0.82±0.06 | 33.85±45.50 | 24.92±22.12 | 43.67±40.39 |
> | **XLarge** | **Optimality Rate (%)** | 100.00±0.00 | 130.83±47.74 | 50.00±61.82 | 5.33±0.00 | 5.33±0.00 | 117.79±158.87 | 5.33±0.00 | 4.92±0.28 | 127.09±168.29 | 93.09±81.28 | 162.97±149.47 |
> | **XLarge** | **Optimality Gap** | 0.00±0.00 | -30.83±47.74 | 50.00±61.82 | 94.67±0.00 | 94.67±0.00 | -17.79±158.87 | 94.67±0.00 | 95.08±0.28 | -27.09±168.29 | 6.91±81.28 | -62.97±149.47 |
>
> While the TARGETNET\E is able to learn a near-optimal policy in one instance of the seed, leading to high performance when the best performing schedule is chosen based on models trained on different seeds, the mean and standard deviation of the performance across seeds show that TARGETNETv2, which removes the softmax of the attention, can not only learn the same policies, but also converge on other locally optimal scheduling policies, making TARGETNETv2 less brittle.
>
> > **Why does sequential version performs better on the largest task? What are the typical failure modes of TARGETNET under this setting?**
>
> We discuss the advantages of sequential version in Line 290. At the extra large scale, the action space for Task Allocation, which is $O(|A||T|)$, becomes as large as 16000, which leads to the probability distribution to get close to uniform, leading to a drop in performance.
>
> [1] Z. Yang, A. Ishay, and J. Lee, “Learning to Solve Constraint Satisfaction Problems with Recurrent Transformer,” Jul. 10, 2023, arXiv:2307.04895. arXiv.2307.04895
>
> [2] Veličković, P., Cucurull, G., Casanova, A., Romero, A., Lio, P., & Bengio, Y. (2017). “Graph attention networks”. arXiv:1710.10903
>
> [3] Hu, Z., Dong, Y., Wang, K., & Sun, Y. (2020, April). “Heterogeneous graph transformer”. In Proceedings of the web conference 2020 (pp. 2704-2710).
>
> [4] Vaswani, A., Shazeer, N., Parmar, N., Uszkoreit, J., Jones, L., Gomez, A. N., ... & Polosukhin, I. (2017). “Attention is all you need.” Advances in neural information processing systems.

---

### Official Review · Reviewer_2QTA · 2025-07-07

**Clarity:** 4
**Significance:** 3
**Originality:** 3
**Rating:** 5
**Confidence:** 4

**Summary:**

* Proposes TARGETNET, an approach which aims to enable the coordination of large teams of heterogenous mobile agents on complex tasks
* TARGETNET uses residual heterogeneous graph transformers, and its task assignment/scheduling capabilities are trained via RL

**Questions:**

* Within the MARL literature, ALMA [1] is one method that may be relevant (it also involves allocating heterogenous agents to tasks dynamically; a learning-based approach); discussion on the differences in these settings, and the comparative merits of TARGETNET and ALMA would be useful. Can TARGETNET be applied in ALMA's setting as well? Or broadly, be integrated with more general, learning-based approaches? (as I reflect, it seems that ALMA's setting is more dynamic than the one investigated here)

[1] https://arxiv.org/abs/2205.14205

**Ethical Concerns:**

["NO or VERY MINOR ethics concerns only"]

**Final Justification:**

I stand by my initial review and therefore maintain my current score.

**Limitations:**

Yes

**Quality:**

4

**Strengths And Weaknesses:**

Strengths
* Problem formulation is clear, and the method is a natural product of reasonable decisions
* The paper presentation is generally very polished
* The evaluation tasks seem reasonable, and the baselines are extensive
* The results are  generally strong, and the following discussion is nuanced and comprehensive

Weaknesses
* None major
* The evaluation is on fairly contrived domains, which seem somewhat removed from real-world settings; related, the authors acknowledge their approach is not directly applicable to more dynamic settings, since the plans are pre-computed
* Nitpicks
    * Figure 2a text is fairly small; would recommend ensuring fontsize in diagrams are generally at the same scale of the caption.

---

> ### Author Rebuttal · Authors · 2025-07-30
>
> We thank the reviewer for their positive assessment and constructive feedback.
>
> > **The evaluation is on fairly contrived domains, which seem somewhat removed from real-world settings; related, the authors acknowledge their approach is not directly applicable to more dynamic settings, since the plans are pre-computed**
>
> Our primary motivation for designing this domain was to fill a specific gap in the literature for multi-scale, heterogeneous multi-agent scheduling in continuous spaces, which requires accounting for agent travel times, task durations, and complex inter-task dependencies. By synthesizing elements from well-established optimization problems like the **Vehicle Routing Problem with Time Windows (VRPTW)** and the **Job Shop Scheduling Problem (JSSP)**, our domain, while simulated, is designed to capture the combinatorial complexity inherent in many real-world logistics and coordination challenges. We will clarify this connection in the paper to better contextualize our environment's relevance.
>
> While the pre-computed plans limit applicability in highly dynamic environments, where frequent replanning is necessary, this assumption is standard for the class of offline task allocation and scheduling problems, including those addressed by our baselines, as well as more standard optimization problems such as **VRP** and **Travelling Salesman Problem (TSP)**. The chosen baselines allows for a focused and fair evaluation of the scheduling algorithm itself, decoupled from the separate and significant challenges of real-time perception and collision avoidance. We believe our work provides a strong foundation for future extensions into more dynamic settings.
>
> To make this clearer, we will expand the limitations section of the revised manuscript to better frame the problem setting and explicitly motivate dynamic re-planning as a key avenue for future work.
>
> > **Nitpicks**
>     - **Figure 2a text is fairly small; would recommend ensuring fontsize in diagrams are generally at the same scale of the caption.**
>
> We modified the Figure 2a to fix the fontsize and further ensured that the font size of the figures match the scale of the captions in the camera ready version.
>
> > **Within the MARL literature, ALMA [1] is one method that may be relevant (it also involves allocating heterogenous agents to tasks dynamically; a learning-based approach); discussion on the differences in these settings, and the comparative merits of TARGETNET and ALMA would be useful. Can TARGETNET be applied in ALMA's setting as well? Or broadly, be integrated with more general, learning-based approaches? (as I reflect, it seems that ALMA's setting is more dynamic than the one investigated here)**
>
> ALMA[1] allows for allocation of agents to task in cooperative tasks followed by a subtask allocator to handle to order of assignments, using a Value-based model. In comparison, TARGETNET is a policy-based framework. While ALMA optimizes for performance of subtask for each subteam, TARGETNET optimizes for the global makespan and feasibility. Furthermore, TARGETNET focuses on generating scheduling prior to any action is taken based on the complete representation of the initial condition.
>
> We will include ALMA[1] into the related section and discuss the differences in application and problem setup.
>
> [1] Iqbal, S., Costales, R., & Sha, F. (2022). Alma: Hierarchical learning for composite multi-agent tasks. _Advances in neural information processing systems_, _35_, 7155-7166.

---

### Decision · Program_Chairs · 2025-09-17

**Decision:**

Accept (poster)

**Comment:**

The paper studies the use of Graph Transformers and RL for certain classes of combinatorial optimization problems. The problem, of finding polynomial time approximately optimal solutions for large-scale combinatorial optimization, is very well-motivated and significant. The proposed solution, TargetNet, is a novel synthesis of techniques from different ML areas. Experiments show that TargetNet is effective in solving several combinatorial problems, with time taken orders of magnitude faster than previous GNN baselines.

Reviewers provided constructive feedback which the authors incorporated during the discussion phase. An important outcome from the additional ablations is that the authors identified a variant, TargetNetv2, that was less brittle than TargetNet across the various problems tested in the experiments. Most of the reviewers agreed that the paper (with the inclusion of TargetNetv2) is above the bar for publication; the only dissent/concern was that including TargetNetv2 in the paper will require all ablations to be re-run with TargetNetv2 as the base, to check if the authors conclusions about the utility of different components of TargetNet remain true of TargetNetv2.